# Entering the Era of Discrete Diffusion Models: A Benchmark for Schrödinger Bridges and Entropic Optimal Transport

**Xavier Aramayo Carrasco**[*][†]
Applied AI Institute,
Moscow, Russia

**Grigoriy Ksenofontov**[*]
Applied AI Institute,
MIRAI[‡],
Moscow, Russia

**Aleksei Leonov**
AI Foundation and Algorithm Lab
MIRAI[‡],
Moscow, Russia

**Iaroslav Koshelev**
AI Foundation and Algorithm Lab
Moscow, Russia

**Alexander Korotin**
Applied AI Institute,
AXXX,
Moscow, Russia

## Abstract

The Entropic Optimal Transport (EOT) problem and its dynamic counterpart, the Schrödinger bridge (SB) problem, play an important role in modern machine learning, linking generative modeling with optimal transport theory. While recent advances in discrete diffusion and flow models have sparked growing interest in applying SB methods to discrete domains, there remains no reliable way to assess how well these methods actually solve the underlying problem. We address this challenge by introducing a benchmark for SB on discrete spaces. Our construction yields pairs of probability distributions with analytically known SB solutions, enabling rigorous evaluation. As a byproduct of building this benchmark, we obtain two new SB algorithms, DLightSB and DLightSB-M, and additionally extend prior related work to construct the $\alpha$-CSBM algorithm. We demonstrate the utility of our benchmark by evaluating both existing and new solvers in high-dimensional discrete settings. This work provides the first step toward proper evaluation of SB methods on discrete spaces, paving the way for more reproducible future studies. The code for the benchmark and all associated experiments is available at
https://github.com/gregkseno/catsbench.

## 1 Introduction

The Entropic Optimal Transport (Cuturi, 2013, EOT) problem and its dynamic counterpart, the Schrödinger bridge (Schrödinger, 1931, SB), have recently attracted significant attention in the machine learning community due to their relevance for generative modeling and unpaired learning. A variety of methods have been developed to solve these problems in *continuous data spaces* such as (Daniels et al., 2021; Gushchin et al., 2023a; 2024b; Mokrov et al., 2024; Vargas et al., 2021; Chen et al., 2021; Shi et al., 2023; De Bortoli et al., 2024; Korotin et al., 2024; Gushchin et al., 2024a).

At the same time, much real world data are *discrete by nature*, including text (Austin et al., 2021; Gat et al., 2024), molecular graphs (Vignac et al., 2022; Qin et al., 2024; Luo et al., 2024), and protein sequences (Campbell et al., 2024). Others are *discrete by construction*, such as vector-quantized representations of images and audio (Van Den Oord et al., 2017; Esser et al., 2021).

Given the prevalence of discrete data and the rapid progress in discrete diffusion/flow models (Hoogeboom et al., 2021; Austin et al., 2021; Campbell et al., 2022; Lou et al., 2023; Sahoo et al., 2024; Shi et al., 2024; Campbell et al., 2024; Gat et al., 2024), research on SBs has attracted attention in recent years. For instance, several works have already taken first steps in this direction (Kim et al., 2024, DDSBM;Ksenofontov & Korotin, 2025, CSBM), adapting diffusion methodologies from (Vignac et al., 2022, DiGress;Austin et al., 2021, D3PM), respectively. Yet, beyond these initial studies, practical, broadly applicable solvers for discrete-space EOT/SB remain largely absent.

---

[*]Equal contribution

[†]Correspondence to: xavier.aramayo2@gmail.com

[‡]Moscow Independent Research Institute of Artificial Intelligence

To make progress on this front, it is also important to have reliable ways to evaluate SB solvers. In practice, EOT/SB methods are often assessed using proxy metrics such as FID (Heusel et al., 2017) or mean-squared-error between input and output. While being useful, these metrics only indirectly reflect whether a method truly solves the EOT/SB problem, since they can be strongly influenced by parameterization, regularization, and other implementation details. Evaluation benchmarks address this limitation by providing a controlled setting in which solvers can be compared against known SB solutions. This allows performance differences to be attributed directly to the underlying algorithms. Consistent with this goal, several optimal transport (OT) benchmarks have been proposed (Korotin et al., 2021; 2022), and similar efforts have recently appeared for continuous-state SB (Gushchin et al., 2023b). However, no analogous benchmark currently exists for discrete data.

From the discussion above, two key limitations in discrete-space EOT/SB research emerge: **(1)** lack of a benchmark to assess the performance of the solvers and **(2)** the limited availability of discrete-space EOT/SB solvers. In this work, we address both issues; our **contributions** are detailed below:

- **Methodology.** We present a general methodology to create pairs of discrete probability distributions with known SB solutions (§3.1). To overcome tractability issues of the methodology in discrete spaces, we introduce a CP-based parameterization (§3.2). This parameterization yields a closed-form SB and enables a practically feasible benchmark construction.

- **Algorithms.** We present various adaptations of continuous-state SB algorithms to discrete settings and evaluate them on our benchmark. Specifically, we develop the following algorithms:

  - DLightSB (§4.3) and DLightSB-M (§4.4) – solvers obtained as a byproduct of the benchmark, mirroring LightSB and LightSB-M (Korotin et al., 2024; Gushchin et al., 2024a);
  - $\alpha$-CSBM (§4.2) – a solver that combines CSBM (Ksenofontov & Korotin, 2025) with the online update strategy of $\alpha$-DSBM (De Bortoli et al., 2024).

**Notation.** We consider a discrete state space $\mathcal{X} = \mathbb{S}^D$, where $\mathbb{S} = \{0, 1, \ldots, S - 1\}$ is the set of $S$ categories and $D$ is the dimensionality. Each $x \in \mathcal{X}$ is a $D$-dimensional vector $x = (x^1, \ldots, x^D)$. Time is discretized as $\{t_n\}_{n=0}^{N+1}$ with $0 = t_0 < t_1 < \ldots < t_N < t_{N+1} = 1$. This gives $N+2$ time points and defines the *path space* $\mathcal{X}^{N+2}$ with the tuple $x_{\text{in}} := (x_{t_1}, \ldots, x_{t_N}) \in \mathcal{X}^N$ collecting the intermediate states. The set $\mathcal{P}(\mathcal{X}^{N+2})$ comprises all discrete time stochastic processes on the path space, with $\mathcal{M}(\mathcal{X}^{N+2}) \subset \mathcal{P}(\mathcal{X}^{N+2})$ denoting the subset of *Markov processes*. Any $q \in \mathcal{M}(\mathcal{X}^{N+2})$ admits forward and backward representations: $q(x_0, x_{\text{in}}, x_1) = q(x_0) \prod_{n=1}^{N+1} q(x_{t_n}|x_{t_{n-1}}) = q(x_1) \prod_{n=1}^{N+1} q(x_{t_{n-1}}|x_{t_n})$. Finally, $q(\cdot|\cdot)$ is used to denote *conditional* $(x_0 \to x_1)$ and *transition* $(x_{t_{n-1}} \to x_{t_n})$ distributions.

## 2 BACKGROUND

This section recalls EOT and SB, which form the core framework for our benchmark and methods. We begin with the dynamic SB formulation and its connection to the static SB problem (§2.1). We then recall different reference processes examples (§2.2) that induce practical cost functions and thereby connect SB to the EOT framework (§2.3). Finally, we introduce the discrete-space generative EOT/SB task and specify the benchmark evaluation criteria used in this work (§2.4).

### 2.1 DYNAMIC AND STATIC SCHRÖDINGER BRIDGES ON DISCRETE SPACES

**Dynamic Schrödinger Bridge.** The original SB problem (Schrödinger, 1931; 1932; Léonard, 2013) seeks to find a process $q^* \in \mathcal{P}(\mathcal{X}^{N+2})$ interpolating between an initial distribution $p_0$ at $t_0 = 0$ and a final distribution $p_1$ at $t_{N+1} = 1$. This distribution is found by minimizing the Kullback-Leibler (KL) divergence with respect to a given *Markov reference process* $q^{\text{ref}} \in \mathcal{M}(\mathcal{X}^{N+2})$ subject to the marginal constraints $p_0(x_0) = q(x_0)$ and $p_1(x_1) = q(x_1)$. One finds the following *optimal process*:

$$q^* = \underset{q \in \Pi_N(p_0, p_1)}{\arg\min} \; \text{KL}\left(q(x_0, x_{\text{in}}, x_1) \| q^{\text{ref}}(x_0, x_{\text{in}}, x_1)\right), \tag{1}$$

where $\Pi_N(p_0, p_1) \subset \mathcal{P}(\mathcal{X}^{N+2})$ denotes the subset of $\mathcal{X}$-valued stochastic processes which have $p_0$ and $p_1$ as marginals at times $t_0 = 0$ and $t_{N+1} = 1$, respectively. In other words, the dynamic SB problem seeks the stochastic process $q^*$ that minimally deviates from a reference process $q^{\text{ref}}$ while respecting the boundary distributions $p_0$ and $p_1$.

**Static Schrödinger Bridge.** The previously introduced dynamic SB problem also allows a static formulation. Linking them begins with observing that (1) admits the following decomposition:

$$\min_{q \in \Pi_N(p_0, p_1)} \Big[ \text{KL}\left(q(x_0, x_1) \| q^{\text{ref}}(x_0, x_1)\right) + \mathbb{E}_{q(x_0, x_1)} \text{KL}\left(q(x_{\text{in}} | x_0, x_1) \| q^{\text{ref}}(x_{\text{in}} | x_0, x_1)\right) \Big]. \quad (2)$$

We further note that the conditional KL term in (2) vanishes when $q(x_{\text{in}} | x_0, x_1) = q^{\text{ref}}(x_{\text{in}} | x_0, x_1)$. Thus, we restrict $q$ to the set of processes that satisfy this condition. This set is known as *the reciprocal class* of $q^{\text{ref}}$ and it is denoted by $\mathcal{R}^{\text{ref}}(\mathcal{X}^{N+2}) \subset \mathcal{P}(\mathcal{X}^{N+2})$. Under this restriction, the optimization reduces to the first KL term alone, leading directly to the static SB problem:

$$q^*(x_0, x_1) = \underset{q \in \Pi(p_0, p_1)}{\arg\min} \; \text{KL}\left(q(x_0, x_1) \| q^{\text{ref}}(x_0, x_1)\right), \quad (3)$$

where $\Pi(p_0, p_1) \subset \mathcal{P}(\mathcal{X}^2)$ denotes the set of joint distributions with marginals $p_0$ and $p_1$, and $q^*(x_0, x_1)$ is *the optimal joint distribution*.

Notably, the static SB formulation is closely related to an EOT problem. This connection is established since the reference process induces a corresponding cost function. To make this link explicit, in the next sections, we first introduce commonly used reference processes on discrete spaces and then derive the corresponding connection between static SB and EOT.

## 2.2 Examples of suitable Reference Processes

The key ingredient in both SB formulations is the Markov reference process $q^{\text{ref}}$. In discrete space, it is typically modeled as a discrete-time Markov chain with strictly positive transitions, i.e., $q^{\text{ref}}(x_{t_n} | x_{t_{n-1}}) > 0$ for all $(x_{t_{n-1}}, x_{t_n})$. As in standard discrete diffusion models (Austin et al., 2021), we focus on factorizable reference processes $q^{\text{ref}}(x_{t_n} | x_{t_{n-1}}) = \prod_{d=1}^{D} q^{\text{ref}}(x_{t_n}^d | x_{t_{n-1}}^d)$ and thus present the construction in the one-dimensional case. We further assume time-homogeneity ($q^{\text{ref}}(x_{t_n}^d | x_{t_{n-1}}^d) = Q^{\text{ref}} \in [0, 1]^{S \times S}$ for all $n$), so that the cumulative transition distributions for $n$-steps are defined by the matrix power $\overline{Q}_n^{\text{ref}} = [Q^{\text{ref}}]^n$.

**Remark.** The reference process $q^{\text{ref}}$ can also be defined in continuous time, where transitions are specified by transition rates (see, e.g., (Campbell et al., 2022)). In this setting, controlling the dynamics is often less direct; moreover, discrete-time Markov chains form a strictly larger class, since not every chain admits a continuous-time analogue (the embeddability problem (Kingman, 1962)). For the convenience of benchmark construction, we therefore focus on the discrete-time setting.

We now introduce two popular diffusion-like transitions: uniform (Hoogeboom et al., 2021; Campbell et al., 2022) and Gaussian-like (Austin et al., 2021).

**Uniform Reference Process ($q^{\text{unif}}$).** For unordered data, where no relation exists between categories, a natural choice is the so-called uniform transition matrix. In this case, for each dimension $d$, the elements of the transition matrix $Q^{\text{ref}}$ are defined by

$$[Q^{\text{ref}}]_{x_{t_{n-1}}^d, x_{t_n}^d} = \begin{cases} 1 - \gamma, & \text{if } x_{t_n}^d = x_{t_{n-1}}^d, \\ \frac{\gamma}{S-1}, & \text{if } x_{t_n}^d \neq x_{t_{n-1}}^d, \end{cases} \quad (4)$$

where $\gamma \in [0, 1]$ is a stochasticity parameter. This reference process assigns equal probability to transitioning into any different category, thereby ignoring any inherent ordering among categories. In Appendix B, we provide a closed-form expression for $q^{\text{ref}}(x_1^d | x_0^d) = \overline{Q}_{N+1}^{\text{ref}}$ in the uniform case.

**Gaussian Reference Process ($q^{\text{gauss}}$).** For ordered data, where categories are expected to exhibit meaningful relations, a Gaussian-like transition matrix is more appropriate. With the stochasticity parameter $\gamma > 0$ and the maximum category distance $\Delta = S - 1$, the transition probabilities are

$$[Q^{\text{ref}}]_{x_{t_{n-1}}^d, x_{t_n}^d} = \frac{\exp\left(-\frac{4(x_{t_n}^d - x_{t_{n-1}}^d)^2}{(\gamma \Delta)^2}\right)}{\sum_{\delta=-\Delta}^{\Delta} \exp\left(-\frac{4\delta^2}{(\gamma \Delta)^2}\right)}, \qquad x_{t_n}^d \neq x_{t_{n-1}}^d. \quad (5)$$

The diagonal entries take the remaining probability so that each row sums to 1.

## 2.3 ENTROPIC OPTIMAL TRANSPORT ON DISCRETE SPACES

Following the construction of the Markov reference process, the static SB problem (§3) takes a form equivalent to the EOT problem (Cuturi, 2013). Concretely, expressing $q^{\text{ref}}(x_0, x_1) = q^{\text{ref}}(x_0)q^{\text{ref}}(x_1|x_0)$ and setting $q^{\text{ref}}(x_0) = p_0(x_0)$ lets us rewrite the minimization in (3) as

$$
\min_{q \in \Pi(p_0,p_1)} \text{KL}\big(q(x_0,x_1)\|q^{\text{ref}}(x_0,x_1)\big) =
$$

$$
= \min_{q \in \Pi(p_0,p_1)} \sum_{x_0,x_1} q(x_0,x_1) \log \frac{q(x_0,x_1)}{q^{\text{ref}}(x_0)q^{\text{ref}}(x_1|x_0)}
$$

$$
= \min_{q \in \Pi(p_0,p_1)} -H(q) - \sum_{x_0,x_1} q(x_0,x_1) \log q^{\text{ref}}(x_1|x_0) \underbrace{-\sum_{x_0,x_1} q(x_0,x_1) \log q^{\text{ref}}(x_0)}_{=-H(p_0)=\text{const}} \quad (6)
$$

$$
= \min_{q \in \Pi(p_0,p_1)} \mathbb{E}_{q(x_0,x_1)}\big[-\log q^{\text{ref}}(x_1|x_0)\big] - H(q) - \text{const}
$$

$$
= \min_{q \in \Pi(p_0,p_1)} \mathbb{E}_{(x_0,x_1)\sim q}\big[c(x_0,x_1)\big] - H(q) - \text{const},
$$

where $H(q)$ is the entropy of $q$, while $H(p_0)$ remains constant when minimizing over $q$. Thus, the static SB formulation becomes equivalent to the entropy-regularized optimal transport problem with cost $c(x_0, x_1) = -\log q^{\text{ref}}(x_1|x_0)$. This establishes a direct correspondence between both SB formulations and EOT, which we collectively referred to as the EOT/SB problem, enabling the construction of a unified benchmark for all three problems.

## 2.4 DISCRETE GENERATIVE PROBLEM SETUP AND EVALUATION PROTOCOL

Building on this background, we formalize the *generative discrete-space EOT/SB task*. This is a well-established problem in the literature (Kim et al., 2024; Ksenofontov & Korotin, 2025). In short, the goal is to learn an optimal conditional distribution that transports a probability distribution on discrete spaces using available empirical samples. Formally, we consider the following setup:

> We assume the learner is given empirical datasets $\{x_0^{(i)}\}_{i \in I_0}$ and $\{x_1^{(j)}\}_{j \in I_1}$, $x_0^{(i)}, x_1^{(j)} \in \mathcal{X}$, consisting of i.i.d. samples from the unknown distributions $p_0, p_1 \in \mathcal{P}(\mathcal{X})$ where $\mathcal{X}$ is a discrete state space. Then, the task is to use these samples to find a solution $q^*$ to the EOT/SB problem (1, 3, 6) between $p_0$ and $p_1$ for a given reference $q^{\text{ref}}$. Moreover, the solution should support out-of-sample generation so that for any new $(x_0^{\text{new}})$ one can generate $x_1^{\text{new}} \sim q^{\text{model}}(x_1|x_0^{\text{new}})$.

Despite recent progress in developing discrete-space SB methods for this task, there is still no standard evaluation protocol. The main obstacle is the lack of discrete datasets with known ground-truth EOT/SB solutions, i.e., pairs of marginals $(p_0, p_1)$ for which the optimal conditional $q^*(x_1|x_0)$ is available. Access to such ground truth enables direct comparison with a learned model $q^{\text{model}}(x_1|x_0)$, allowing evaluation of how accurately a method solves the underlying EOT/SB problem rather than relying on proxy metrics. Inspired by (Gushchin et al., 2023b), we therefore propose a pipeline that generates ground-truth benchmark instances, applicable to discrete-space solvers.

**Remark.** Our paper is not related to the discrete EOT, which includes solvers such as the Sinkhorn algorithm (Cuturi, 2013) or gradient-based methods (Dvurechensky et al., 2018). These approaches are designed for a non-generative problem setting, see (Ksenofontov & Korotin, 2025, §2.3). They treat samples as empirical distributions $p_0(x_0) = \frac{1}{|I_0|}\sum_{i \in I_0} \delta_{x_0^{(i)}}$, $p_1(x_1) = \frac{1}{|I_1|}\sum_{j \in I_1} \delta_{x_1^{(j)}}$. The resulting joint distribution is then a bi-stochastic $|I_0| \times |I_1|$ matrix, which does not support out-of-sample generation. While some extensions attempt to provide inference for unseen data (Hütter & Rigollet, 2021; Pooladian & Niles-Weed, 2021; Manole et al., 2024; Deb et al., 2021), they are designed for continuous spaces ($\mathcal{X} = \mathbb{R}^D$) rather than the discrete ($\mathcal{X} = \mathbb{S}^D$) considered in our work.

## 3 BENCHMARK

In this section, we address the absence of evaluation benchmarks for discrete-space EOT/SB solvers by proposing a novel benchmark construction. In §3.1 we present the theoretical foundations of the construction. Next, we make it tractable via a CP parameterization in §3.2. Finally, we use this parameterization to construct a high-dimensional Gaussian mixture benchmark in §3.3. We provide detailed proofs for all stated theoretical results in Appendix A.

### 3.1 MAIN THEOREM FOR BENCHMARK CONSTRUCTION

For an initial distribution $p_0 \in \mathcal{P}(\mathcal{X})$, we aim to construct a target distribution $p_1 \in \mathcal{P}(\mathcal{X})$ such that the optimal joint distribution $q^*(x_0, x_1)$ between them is known by our construction. The resulting pair $(p_0, p_1)$ together with $q^*$ can then be used as benchmark data for evaluating SB methods. Our following theorem plays the key role in the construction of benchmark pairs.

**Theorem 3.1** (Benchmark Pair Construction for Discrete-Space EOT/SB). *Let $p_0 \in \mathcal{P}(\mathcal{X})$ be a given initial distribution on a discrete space $\mathcal{X}$ and $v^* : \mathcal{X} \rightarrow \mathbb{R}$ be a given scalar-valued function. Consider a joint distribution $q^* \in \mathcal{P}(\mathcal{X}^2)$ such that $q^*(x_0) = p_0(x_0)$ and $q^*(x_1|x_0) \propto v^*(x_1)q^{\text{ref}}(x_1|x_0)$ define $p_1(x_1) := q^*(x_1)$ as its second marginal. Then $q^*$ together with the reference process $q^{\text{ref}}$ defines the discrete-space EOT/SB (1,3,6) between $p_0$ and $p_1$.*

Similar results in continuous spaces appear in (Gushchin et al., 2023b). In Theorem 3.1 we provide a discrete-space analog which shows that any pair $(p_0, v^*)$ induces a corresponding pair $(p_0, p_1)$ with a closed-form $q^*(x_1|x_0)$ on **discrete space**. We refer to the latter pair as a *benchmark pair*. However, this construction specifies $q^*(x_1|x_0)$ only up to proportionality, necessitating the normalized form:

$$q^*(x_1|x_0) = \tfrac{1}{c^*(x_0)} v^*(x_1) q^{\text{ref}}(x_1|x_0), \tag{7}$$

where $c^*(x_0) := \sum_{x_1 \in \mathcal{X}} v^*(x_1)q^{\text{ref}}(x_1|x_0)$ is the normalization constant. Although this provides a closed-form expression for $q^*$, implementing it in high-dimensional spaces ($|\mathcal{X}| = S^D$) remains computationally challenging. In particular, evaluating the normalizing constant and sampling from $q^*$ are non-trivial tasks. To address these challenges, we introduce a CP-parameterization that allows efficient computation and sampling, as detailed in the next section.

### 3.2 PRACTICAL PARAMETERIZATION

We parameterize the scalar-valued function $v^*$ using a rank-1 Canonical Polyadic (CP) decomposition, which captures interactions across dimensions and provides a compact yet expressive representation. Such decompositions act as universal approximators, capable of modeling complex functions when the rank is sufficiently large (Cohen et al., 2016; Basharin et al., 2025). Thus, $v^*$ is written as

$$v^*(x_1) = \sum_{k=1}^{K} \beta_k \prod_{d=1}^{D} r_k^d[x_1^d]. \tag{8}$$

Expression (8) defines a mixture of $K$ *factorizable distributions*, each with weight $\beta_k \geq 0$. For each mixture component $k$ and dimension $d$, probabilities are defined by non-negative vectors $r_k^d \in \mathbb{R}_+^S$, referred to as *CP cores*, where $r_k^d[x_1^d]$ denotes the probability of state $x_1^d$.

**Proposition 3.1** (Tractable Parameterization of Conditional Distributions). *Let $q^{\text{ref}}$ be a factorizable Markov reference process on a discrete space $\mathcal{X}$. Using the CP decomposition of the scalar-valued function $v^*$ in (8), the optimal conditional distribution satisfies:*

$$q^*(x_1|x_0) = \qquad\qquad\qquad\qquad c^*(x_0) =$$

$$\frac{1}{c^*(x_0)} \sum_{k=1}^{K} \beta_k \prod_{d=1}^{D} \Big[ r_k^d[x_1^d] q^{\text{ref}}(x_1^d|x_0^d) \Big]; \quad (9) \qquad \sum_{k=1}^{K} \beta_k \prod_{d=1}^{D} \left( \sum_{x_1^d=0}^{S-1} r_k^d[x_1^d] q^{\text{ref}}(x_1^d|x_0^d) \right) \quad (10)$$

*where $c^*(x_0)$ is the normalization constant. This formulation expresses $q^*(x_1|x_0)$ as a mixture of $K$ factorizable distributions, each weighted by a scalar coefficient $\beta_k$.*

A key consequence of Proposition 3.1 is that $c^*(x_0)$ and $q^*(x_1|x_0)$ are computationally tractable, since both expressions factorize into products of one-dimensional sums. In particular, instead of summing over the full joint space of size $S^D$, computing $c^*(x_0)$ requires only $K$ products of $D$ scalar sums over $S$ states, reducing the complexity from $\mathcal{O}(S^D)$ to $\mathcal{O}(KDS)$. The evaluation of $q^*(x_1|x_0)$ is similar. In practice, computations are performed using log-sum-exp operations for numerical stability. To sample from $q^*(x_1|x_0)$, we use ancestral sampling: we first choose a mixture component $k$ with probability proportional to $\beta_k \prod_{d=1}^{D} \sum_{x_1^d} r_k^d[x_1^d] q^{\text{ref}}(x_1^d \mid x_0^d)$, and then, conditioned on $k$, sample each coordinate $x_1^d$ independently from $q^*(x_1^d \mid x_0^d) \propto r_k^d[x_1^d] q^{\text{ref}}(x_1^d \mid x_0^d)$.

Previously, we described the benchmark construction in the static SB (equivalently, EOT) setting, with focus on $q^*(x_1|x_0)$. Nevertheless, a similar result can be obtained for the dynamic SB and its forward Markov representation defined by transition distributions $q^*(x_{t_n}|x_{t_{n-1}})$. More precisely, these transitions can be obtained by reweighting the reference process transitions $q^{\text{ref}}(x_{t_n}|x_{t_{n-1}})$ with time-dependent scalar-valued functions $\phi_t^*$ (Georgiou & Pavon, 2015, Thm. 2):

$$q^*(x_{t_n}|x_{t_{n-1}}) = q^{\text{ref}}\left(x_{t_n}|x_{t_{n-1}}\right) \frac{\phi_{t_n}^*(x_{t_n})}{\phi_{t_{n-1}}^*(x_{t_{n-1}})}, \qquad \phi_{t_n}^*(x_{t_n}) = \mathbb{E}_{q^{\text{ref}}(x_1|x_{t_n})}\left[v^*(x_1)\right]. \quad (11)$$

The next proposition provides a tractable form of the corresponding transition distributions.

**Proposition 3.2** (Tractable Parameterization of Conditional SB Transition Distributions). *Let $q^{\text{ref}}$ be a factorizable Markov reference process on a discrete space $\mathcal{X}$. Using the CP decomposition of the scalar-valued function $v^*$ in (8) together with the definition of the time-dependent functions $\phi_{t_n}^*$ in (11), the optimal transition distributions satisfy:*

$$q^*(x_{t_n}|x_{t_{n-1}}) \propto q^{\text{ref}}(x_{t_n}|x_{t_{n-1}}) \sum_{k=1}^{K} \beta_k \prod_{d=1}^{D} u_{k,t_n}^d[x_{t_n}^d]. \quad (12)$$

*where $u_{k,t_n}^d\left[x_{t_n}^d\right] = \sum_{x_1^d=0}^{S-1} q^{\text{ref}}(x_1^d|x_{t_n}^d)r_k^d\left[x_1^d\right]$. Sampling is done via ancestral sampling.*

By the same argument as for $q^*(x_1|x_0)$, the transition distributions $q^*(x_{t_n}|x_{t_{n-1}})$ are tractable. In practice, normalization of (12) is carried out in the log-domain by subtracting the log-sum-exp over all states to obtain numerically stable transition probabilities.

### 3.3 HIGH-DIMENSIONAL GAUSSIAN MIXTURES BENCHMARK CONSTRUCTION

We now instantiate the proposed construction. We set $p_0$ as a discretized Gaussian on $D \in \{2, 16, 64\}$ dimensions with $S = 50$ categories. For $v^*$, we use $K = 4$ components with uniformly initialized weights $\beta \in \mathbb{R}^K$. The CP cores are initialized using discretized Gaussian probabilities, with means uniformly sampled on a sphere of radius 5 and a standard deviation fixed to $\{1.5, 1.5, 2.5\}$ for the lowest to highest dimensions, respectively. Given $p_0$ and $v^*$, we then construct $p_1$. This initialization produces a target $p_1$ resembling a discretized Gaussian mixture with a clear visual structure. We construct pairs under different reference processes: $q^{\text{gauss}}$ with $\gamma \in \{0.02, 0.05\}$ and $q^{\text{unif}}$ with $\gamma \in \{0.005, 0.01\}$, using $N + 1 = 128$ for both. Figure 1b shows the resulting benchmark pairs.

## 4 SOLVERS FOR EVALUATION

In this section, we recall available discrete EOT/SB solvers as well as addressing the limited availability of such approaches. We begin by recalling *Categorical Schrödinger Bridge Matching (CSBM)* (Ksenofontov & Korotin, 2025), which is an existing SB method tailored to categorical distributions. We then introduce *α-CSBM*, which incorporates the online update strategy of (De Bortoli et al., 2024) into the CSBM framework. Next, we propose *Discrete Light Schrödinger Bridge (DLightSB)* obtained as a byproduct of our benchmark construction (§3) and extending (Korotin et al., 2024) to the discrete setting. Finally, we present *Discrete Light Schrödinger Bridge Matching (DLightSB-M)*, a dynamic extension of DLightSB following (Gushchin et al., 2024a).

### 4.1 CATEGORICAL SCHRÖDINGER BRIDGE MATCHING (CSBM)

In (Ksenofontov & Korotin, 2025, Theorem 3.1), the discrete-space dynamic SB problem is addressed by the *discrete-time Iterative Markovian Fitting (D-IMF) procedure*, whose convergence is established by extending the discrete-time existence theorem of (Gushchin et al., 2024b, Theorem 3.6) to the discrete space and time setting. The exponential convergence rates of D-IMF can be found in (Sokolov & Korotin, 2025). This constructive method uses the fact that the dynamic SB $q^*$ is both reciprocal ($q^* \in \mathcal{R}^{\text{ref}}(\mathcal{X}^{N+2})$) and Markov ($q^* \in \mathcal{M}(\mathcal{X}^{N+2})$). The D-IMF algorithm alternates between projections onto these two sets, starting from an initial process $q^0(x_0, x_1)q^{\text{ref}}(x_{\text{in}}|x_0, x_1)$, where $q^0(x_0, x_1) \in \Pi(p_0, p_1)$, e.g., $p_0(x_0)p_1(x_1)$, and converges to the SB $q^*$ in KL. Namely,

$$q^{2l(+2)} \underset{\text{proj}_{\mathcal{R}^{\text{ref}}}}{\overset{\text{proj}_{\mathcal{M}}}{\rightleftarrows}} q^{2l+1} \quad l = 0, 1, \dots$$

where

$$[\text{proj}_{\mathcal{R}^{\text{ref}}}(q)](x_0, x_{\text{in}}, x_1) = q(x_0, x_1)q^{\text{ref}}(x_{\text{in}}|x_0, x_1), \quad \forall q \in \mathcal{P}(\mathcal{X}^{N+2}), \quad (13)$$

$$[\text{proj}_{\mathcal{M}}(q)](x_0, x_{\text{in}}, x_1) = \underset{m \in \mathcal{M}(\mathcal{X}^{N+2})}{\arg\min} \text{KL}\left(q(x_0, x_{\text{in}}, x_1) \| m(x_0, x_{\text{in}}, x_1)\right), \quad \forall q \in \mathcal{R}^{\text{ref}}(\mathcal{X}^{N+2}). \quad (14)$$

**Loss.** Fortunately, the reciprocal part (13) is straightforward via ancestral sampling. In turn, to make Markovian step (14) tractable, the authors parameterize the transitions of $m$, writing $m = q_\theta$, and minimize the following objective, defined up to an additive constant independent of $\theta$:

$$\mathcal{L}(\theta) = \mathbb{E}_{q(x_0, x_1)} \left[ \sum_{n=1}^{N} \mathbb{E}_{q^{\text{ref}}(x_{t_{n-1}}|x_0, x_1)} \left[ \text{KL}\left( q^{\text{ref}}(x_{t_n}|x_{t_{n-1}}, x_1) \| q_\theta(x_{t_n}|x_{t_{n-1}}) \right) \right] - \right.$$
$$\left. - \mathbb{E}_{q^{\text{ref}}(x_{t_N}|x_0, x_1)}[\log q_\theta(x_1|x_{t_N})] \right]. \quad (15)$$

In practice, the D-IMF procedure is implemented in a bidirectional manner (see (Ksenofontov & Korotin, 2025, §3.2.5)). That is, the forward and backward models are trained alternately at each Markovian step. Notably, the KL loss can be replaced by any divergence from the Bregman family (e.g., the mean squared error (MSE)), introducing an additional hyperparameter in our experimental setup. For details on this equivalence, see (Ksenofontov & Korotin, 2025, App. C.1).

**Remark.** A continuous-time IMF was introduced in the Discrete Diffusion Schrödinger Bridge Matching (Kim et al., 2024, DDSBM) paper, which performs the Markovian projection (14) by matching the generator matrices of continuous-time Markov chains. As it reduces to the same loss and inference process due to the necessity to discretize time, we report results only for CSBM.

## 4.2 $\alpha$-Categorical Schrödinger Bridge Matching ($\alpha$-CSBM)

Running the IMF procedure bidirectionally is often beneficial, see (Kholkin et al., 2026), but it doubles the computational burden, since two neural networks must be trained to model the forward and backward representations. To slightly mitigate this cost, recent work has proposed an online alternative to IMF, called $\alpha$-IMF (De Bortoli et al., 2024; Peluchetti, 2025).

In this approach, the exact projections in (13) and (14) are replaced by partial updates (De Bortoli et al., 2024, Eq. 9). Concretely, rather than running each projection to full convergence, one performs a single optimization step at each iteration $l$. Although each update is incomplete, the alternating steps still steer the learned distribution toward the double projection $\text{proj}_{\mathcal{R}^{\text{ref}}}(\text{proj}_{\mathcal{M}}(\cdot))$, as in IMF. Inspired by these advances in the continuous setting, we extend the same ideas to CSBM (§4.1), viewing the discrete version of $\alpha$-IMF as a heuristic analogue of the original procedure.

**Loss.** Since the method does not require each projection to fully converge, we can take a single optimization step for both representation directions simultaneously. This allows us to extend the CSBM bidirectional setup (§4.1) by jointly updating models using objective (15) in both directions:

$$\mathcal{L}(\theta) = \tfrac{1}{2}\left( \text{KL}\left(\overrightarrow{r_{\text{sg}}}(x_0, x_{\text{in}}, x_1) \| \overleftarrow{q_\theta}(x_0, x_{\text{in}}, x_1)\right) + \text{KL}\left(\overleftarrow{r_{\text{sg}}}(x_0, x_{\text{in}}, x_1) \| \overrightarrow{q_\theta}(x_0, x_{\text{in}}, x_1)\right) \right), \quad (16)$$

where $\rightarrow$ and $\leftarrow$ denote the direction of the Markov representation (typically implemented by conditioning a neural network on a direction variable), and $r_{\text{sg}}$ denotes $\text{proj}_{\mathcal{R}^{\text{ref}}}(q_\theta)$ with stop-gradient.

**Limitation.** To avoid evaluating the full space of size $S^D$, transition probabilities $q_\theta(x_{t_n}|x_{t_{n-1}})$ are factorized across dimensions, reducing it to $D \times S$. However, this parametrization constitutes a key limitation of ($\alpha$-)CSBM, as it introduces approximation error.

## 4.3 Discrete Light Schrödinger Bridge (DLightSB)

Now we take a different direction and propose new solvers for discrete-space EOT/SB problems. Specifically, we focus on methods that arise directly as a byproduct of our benchmark construction. We begin by introducing a new static SB solver, which we entitle DLightSB. In what follows, we adopt the same CP benchmark parameterization (§3.2) for the conditional distribution $q(x_1|x_0)$. In particular, we treat the weights $\beta_k$ and the CP cores $r_k^d$ as learnable parameters and collect them in $\theta = \{\beta_k, r_k^d\}$, yielding the model $q_\theta(x_1|x_0)$ with scalar-valued function $v_\theta(x_1)$.

**Loss.** To optimize the parameters $\theta$, we follow the approach of Korotin et al. (2024). Concretely, we derive a discrete objective that is equivalent to the direct KL objective $\text{KL}\left(q^*\|q_\theta\right)$ up to an additive constant. Importantly, this reformulation removes the dependence on the unknown optimal joint distribution $q^*$ and enables a feasible optimization procedure, as shown in the following proposition.

**Proposition 4.1** (Feasible Discrete Reformulation of the Direct KL Objective). *Under the parametrization (9) of $q_\theta(x_1|x_0)$, the objective $KL\left(q^*\|q_\theta\right)$ admits the following reformulation:*

$$KL\left(q^*\|q_\theta\right) = \mathcal{L}(\theta) - \mathcal{L}^*, \text{ where}$$

$$\mathcal{L}(\theta) = \mathbb{E}_{p_0(x_0)}\big[\log c_\theta(x_0)\big] - \mathbb{E}_{p_1(x_1)}\big[\log v_\theta(x_1)\big], \tag{17}$$

*and $\mathcal{L}^* \in \mathbb{R}$ is a constant value not depending on $\theta$, therefore, it can be omitted.*

Notably, the expectations in Proposition 4.1 can be efficiently estimated via Monte Carlo sampling, and the resulting objective can be optimized using stochastic gradient descent with respect to $\theta$.

### 4.4 DISCRETE LIGHT SCHRÖDINGER BRIDGE MATCHING (DLIGHTSB-M)

Finally, we introduce DLightSB-M, a dynamic variant of DLightSB, which also uses our benchmark construction. Inspired by (Gushchin et al., 2024a), we introduce a *discrete-space optimal projection* that recovers the SB through a single projection step. In particular, rather than projecting a reciprocal process $r \in \mathcal{R}^{\text{ref}}(\mathcal{X}^{N+2})$ onto the Markov set $\mathcal{M}(\mathcal{X}^{N+2})$ as in the D-IMF procedure (14), we directly project $r$ onto the set of all SBs, defined as follows:

$$\mathcal{S}(\mathcal{X}^{N+2}) \coloneqq \Big\{ q^{\text{SB}} \in \mathcal{P}(\mathcal{X}^{N+2}) \text{ such that } \exists\, q_0^{\text{SB}}, q_1^{\text{SB}} \in \mathcal{P}(\mathcal{X}),$$

$$q^{\text{SB}} = \underset{q\in\Pi_N(q_0^{\text{SB}}, q_1^{\text{SB}})}{\arg\min} \; \text{KL}\left(q\|q^{\text{ref}}\right) \Big\}. \tag{18}$$

To justify this projection in discrete settings, we show that (Gushchin et al., 2024a, Theorem 3.1) extends to an arbitrary Markov reference process $q^{\text{ref}}$.

**Proposition 4.2** (Discrete-Space Optimal Projection with an Arbitrary Reference Process). *Let $r \in \mathcal{R}^{\text{ref}}(\mathcal{X}^{N+2})$ be a reciprocal process defined with a reference process $q^{\text{ref}} \in \mathcal{M}(\mathcal{X}^{N+2})$ and a joint distribution $r(x_0, x_1) \in \Pi(p_0, p_1)$. Then, the optimal projection of $r$ onto the set of all SBs $\mathcal{S}(\mathcal{X}^{N+2})$ is the SB $q^*$ between the desired marginals $p_0$ and $p_1$, i.e.,*

$$q^* = \underset{q^{\text{SB}}\in\mathcal{S}(\mathcal{X}^{N+2})}{\arg\min} \; KL\left(r\|q^{\text{SB}}\right). \tag{19}$$

**Loss.** To complete the construction, we need to parameterize $q^{\text{SB}}$ so that the minimization is restricted to $\mathcal{S}(\mathcal{X}^{N+2})$. To address this we recall that $q^{\text{SB}} \in \mathcal{M}(\mathcal{X}^{N+2})$ (§4.1) and it is fully determined by its forward transitions $q(x_{t_n}|x_{t_{n-1}})$. We therefore adopt the closed-form transition parameterization (12), with parameters $\theta = \{\beta_k, r_k^d\}$ as in DLightSB, yielding the transitions $q_\theta(x_{t_n}|x_{t_{n-1}})$ that define an SB. Finally, under this parameterization, the discrete-space optimal projection (19) can be optimized using the Markovian projection objective (15) via stochastic gradient descent.

Importantly, since the reference process $q^{\text{ref}}$ factorizes across dimensions, the KL and cross-entropy (CE) terms in (15) admit an equivalent reformulation as a sum over dimensions. Concretely, each term reduces to a sum over dimensions of KL or CE between $q^{\text{ref}}(x_{t_n}^d|x_{t_{n-1}}^d, x_1^d)$ and the model's $d$-th transition. The latter is obtained by summing out all dimensions except the $d$-th one (see (34) in the proof of Proposition 3.2), which is tractable under our parameterization.

**Limitation.** In high dimensions, the CP parameterization requires a large number of components $K$ to capture complex structure. This makes DLightSB(-M) methods computationally demanding.

## 5 EVALUATION

In this section, we first introduce the evaluation metrics and describe the baselines (§5.1). We then evaluate the solvers from §4 on our benchmark setup (§3.3) and report the results in §5.2.

### 5.1 METRICS AND BASELINES

Evaluating generative models on discrete data is challenging because widely used metrics, such as generative perplexity for text or FID for images (Heusel et al., 2017), are domain-specific. Following tabular-data evaluation work (Zhang et al., 2024; Shi et al., 2025), we adopt the *Shape Score* and *Trend Score* metrics. Further, we introduce an EOT/SB-specific metric given by the *Trajectory KL*.

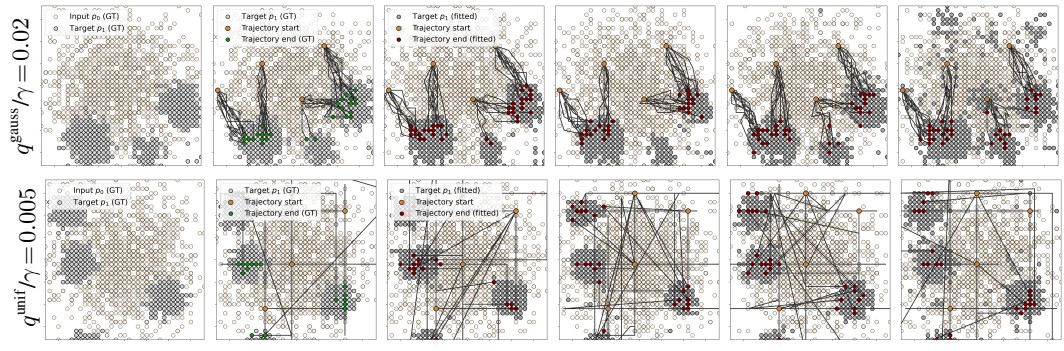

(a) Input/Target     (b) Benchmark     (c) CSBM     (d) $\alpha$-CSBM     (e) DLightSB     (f) DLightSB-M

Figure 1: Samples from all considered methods on our high-dimensional Gaussian mixture benchmark. **Top row**: $q^{\text{unif}}$ ($\gamma = 0.005$). **Bottom row**: Gaussian benchmark ($\gamma = 0.02$). CSBM, $\alpha$-CSBM, and DLightSB-M were trained with KL loss ($N{+}1 = 64$).

**Shape and Trend Score.** These metrics quantify how well a solver matches the benchmark at the level of dimension-wise and pair-wise marginals. For each coordinate $d$ (Shape) and each pair $(d_m, d_n)$ (Trend), we compute an adjusted total variation score and then average over dimensions:

$$\text{SSM}^d = \qquad\qquad\qquad\qquad\qquad \text{TSM}^{d_m, d_n} =$$
$$1 - \frac{1}{2}\sum_{x^d = 0}^{S-1} |\tilde{q}^*(x^d) - \tilde{q}_\theta(x^d)|; \qquad 1 - \frac{1}{2}\sum_{x^{d_m}=0}^{S-1}\sum_{x^{d_n}=0}^{S-1} |\tilde{q}^*(x^{d_m}, x^{d_n}) - \tilde{q}_\theta(x^{d_m}, x^{d_n})|,$$

where the tilde denotes empirical distribution, i.e., $\tilde{q}(x) = \frac{1}{|I|}\sum_{i \in I} \delta_{x^{(i)}}$.

**Conditional Metrics.** In our evaluation, we primarily report conditional variants of these metrics. These are computed by averaging SSM and TSM over sampled $x_0 \sim p_0$ and multiple generated $x_1$ for each $x_0$. This approach provides a direct measure of the fidelity of the learned conditional distribution $q_\theta(x_1|x_0)$ and quantifies how well the EOT/SB solver solves the underlying problem.

**Trajectory KL divergence.** Additionally, we utilize the dynamic SB and its Markov property to compute the forward and reverse KL divergences between processes, namely $\text{KL}\left(q^*(x_0, x_{\text{in}}, x_1)\|q_\theta(x_0, x_{\text{in}}, x_1)\right)$ and $\text{KL}\left(q_\theta(x_0, x_{\text{in}}, x_1)\|q^*(x_0, x_{\text{in}}, x_1)\right)$, respectively, i.e.,

$$\text{KL}\left(q^*(x_0, x_{\text{in}}, x_1)\|q_\theta(x_0, x_{\text{in}}, x_1)\right) = \sum_{n=1}^{N+1}\sum_{d=1}^{D} \text{KL}\left(q^*(x_{t_n}^d|x_{t_{n-1}})\|q_\theta(x_{t_n}^d|x_{t_{n-1}})\right).$$

An analogous decomposition holds for the reverse KL divergence. This metric relies on factorization across dimensions specifically for methods that model only marginals (e.g., CSBM, §4.1). In contrast, for methods that do not impose such a restriction (e.g., DLightSB, §4.3), the required transitions are obtained by marginalizing out all coordinates except the $d$-th dimension, see (34).

**Baselines.** To provide a simple point of reference for our benchmark, we consider three simple baselines. The first, *Independent*, ignores the joint distribution between $x_0$ and $x_1$ and samples $x_1$ directly from $p_1$. The second, *Reference*, takes $x_1$ to be a sample from the reference process $q^{\text{ref}}$. The third, *Feature-wise SB*, solves an SB problem separately for each dimension using the analytical D-IMF procedure (Ksenofontov & Korotin, 2025, Section 4.1) with empirically estimated marginals $p_0$ and $p_1$. This baseline assumes factorization across dimensions and samples each $x_1^d$ independently.

## 5.2 RESULTS

**Benchmark Usage Protocol.** To simulate the generative discrete-space EOT/SB task discussed in §2.4, we use our benchmark pair constructor differently for training and testing. For *training*, we sample $x_0^{\text{train}} \sim p_0$ and $x_1^{\text{train}} \sim p_1$ independently. We further remove dataset-size limitations that could affect a direct comparison of methods by infinitely sampling the training data, as enabled by our benchmark. For *testing*, unconditional metrics and the Trajectory KL are computed on a fixed set of 20 000 precomputed benchmark pairs $(x_0, x_1) \sim q^*(x_0, x_1)$. Conditional metrics are evaluated on 1 000 $x_1$ samples for each of 157 unique $x_0$, where the $x_0$ are drawn from this 20 000 set and the corresponding $x_1$ are sampled from both the method and the benchmark. We also compare

| | | | D=2 | | | | D=16 | | | | D=64 | | | | D=2 | | | | D=16 | | | | D=64 | | | |
| | | | gaussian | | uniform | | gaussian | | uniform | | gaussian | | uniform | | gaussian | | uniform | | gaussian | | uniform | | gaussian | | uniform | |
| Method | Loss | N+1 | 0.02 | 0.05 | 0.005 | 0.01 | 0.02 | 0.05 | 0.005 | 0.01 | 0.02 | 0.05 | 0.005 | 0.01 | 0.02 | 0.05 | 0.005 | 0.01 | 0.02 | 0.05 | 0.005 | 0.01 | 0.02 | 0.05 | 0.005 | 0.01 |
|---|---|---|---|---|---|---|---|---|---|---|---|---|---|---|---|---|---|---|---|---|---|---|---|---|---|---|
| *Independent* | – | – | 0.51 | 0.83 | 0.63 | 0.64 | 0.70 | 0.74 | 0.50 | 0.57 | 0.65 | 0.66 | 0.54 | 0.61 | 0.47 | 0.78 | 0.52 | 0.55 | 0.57 | 0.68 | 0.37 | 0.46 | 0.48 | 0.51 | 0.35 | 0.43 |
| *Reference* | – | – | 0.17 | 0.45 | 0.39 | 0.42 | 0.29 | 0.29 | 0.30 | 0.33 | 0.41 | 0.37 | 0.36 | 0.35 | 0.09 | 0.28 | 0.23 | 0.27 | 0.11 | 0.08 | 0.09 | 0.10 | 0.20 | 0.12 | 0.14 | 0.11 |
| *Feature-wise SB* | – | – | 0.71 | 0.87 | 0.88 | 0.89 | 0.81 | 0.76 | 0.61 | 0.67 | 0.93 | 0.66 | 0.73 | 0.70 | 0.59 | 0.63 | 0.70 | 0.74 | 0.69 | 0.53 | 0.35 | 0.37 | 0.84 | 0.40 | 0.53 | 0.46 |
| DLightSB | – | – | **0.95** | **0.92** | **0.93** | **0.93** | 0.85 | **0.93** | 0.92 | **0.93** | **0.94** | **0.91** | **0.93** | **0.92** | **0.91** | **0.85** | **0.87** | **0.87** | 0.76 | **0.84** | **0.84** | **0.84** | **0.85** | **0.75** | **0.82** | **0.77** |
| CSBM | KL | 16 | 0.72 | 0.71 | 0.87 | 0.88 | 0.79 | 0.74 | 0.64 | 0.65 | 0.86 | 0.85 | 0.65 | 0.68 | 0.64 | 0.61 | 0.80 | 0.81 | 0.67 | 0.61 | 0.52 | 0.51 | 0.76 | 0.69 | 0.52 | 0.54 |
| | | 64 | 0.88 | 0.87 | 0.90 | 0.92 | 0.87 | 0.82 | 0.67 | 0.71 | 0.89 | 0.85 | 0.62 | 0.68 | 0.84 | 0.79 | 0.84 | 0.85 | 0.80 | 0.74 | 0.57 | 0.62 | 0.79 | 0.70 | 0.49 | 0.54 |
| | MSE | 16 | 0.48 | 0.71 | 0.78 | 0.82 | 0.77 | 0.71 | 0.52 | 0.55 | 0.73 | 0.80 | 0.59 | 0.62 | 0.44 | 0.63 | 0.69 | 0.73 | 0.66 | 0.57 | 0.39 | 0.42 | 0.59 | 0.66 | 0.45 | 0.46 |
| | | 64 | 0.29 | 0.82 | 0.77 | 0.77 | 0.82 | 0.81 | 0.59 | 0.64 | 0.66 | 0.87 | 0.62 | 0.63 | 0.23 | 0.75 | 0.69 | 0.67 | 0.73 | 0.72 | 0.48 | 0.55 | 0.50 | 0.71 | 0.48 | 0.48 |
| α-CSBM | KL | 16 | 0.66 | 0.73 | 0.89 | 0.89 | 0.72 | 0.79 | 0.66 | 0.68 | 0.87 | 0.81 | 0.63 | 0.67 | 0.59 | 0.62 | 0.82 | 0.81 | 0.61 | 0.66 | 0.55 | 0.56 | 0.77 | 0.66 | 0.48 | 0.52 |
| | | 64 | 0.77 | 0.87 | 0.90 | 0.91 | 0.89 | 0.88 | 0.66 | 0.73 | 0.89 | 0.83 | 0.63 | 0.66 | 0.73 | 0.79 | 0.83 | 0.84 | 0.82 | 0.80 | 0.56 | 0.64 | 0.79 | 0.68 | 0.49 | 0.51 |
| | MSE | 16 | 0.53 | 0.71 | 0.85 | 0.87 | 0.72 | 0.78 | 0.57 | 0.63 | 0.74 | 0.77 | 0.60 | 0.62 | 0.47 | 0.61 | 0.77 | 0.78 | 0.60 | 0.65 | 0.45 | 0.51 | 0.59 | 0.62 | 0.45 | 0.46 |
| | | 64 | 0.88 | 0.86 | 0.84 | 0.88 | 0.70 | 0.84 | 0.60 | 0.66 | 0.65 | 0.84 | 0.61 | 0.64 | 0.84 | 0.78 | 0.76 | 0.79 | 0.59 | 0.76 | 0.49 | 0.56 | 0.49 | 0.69 | 0.47 | 0.49 |
| DLightSB-M | KL | 16 | 0.86 | 0.91 | 0.92 | 0.93 | 0.81 | 0.91 | 0.92 | 0.92 | 0.83 | 0.76 | 0.68 | 0.73 | 0.82 | 0.85 | 0.86 | 0.86 | 0.70 | 0.82 | 0.83 | 0.82 | 0.73 | 0.62 | 0.47 | 0.50 |
| | | 64 | 0.85 | 0.91 | 0.92 | 0.92 | 0.87 | 0.92 | 0.92 | 0.92 | 0.83 | 0.85 | 0.70 | 0.84 | 0.80 | 0.84 | 0.85 | 0.86 | 0.79 | 0.83 | 0.83 | 0.82 | 0.73 | 0.71 | 0.49 | 0.67 |
| | MSE | 16 | 0.71 | 0.91 | 0.82 | 0.90 | 0.62 | 0.92 | 0.87 | 0.88 | 0.68 | 0.83 | 0.78 | 0.60 | 0.62 | 0.84 | 0.76 | 0.84 | 0.45 | 0.82 | 0.78 | 0.79 | 0.53 | 0.68 | 0.65 | 0.33 |
| | | 64 | 0.69 | 0.91 | 0.79 | 0.89 | 0.58 | 0.92 | 0.87 | 0.85 | 0.57 | 0.83 | 0.53 | 0.76 | 0.60 | 0.83 | 0.72 | 0.82 | 0.40 | 0.81 | 0.79 | 0.76 | 0.39 | 0.68 | 0.33 | 0.60 |

(a) Conditional Shape Score (↑)  (b) Conditional Trend Score (↑)

Table 1: Conditional metrics on our high-dimensional Gaussian mixture benchmark. The best-performing method is highlighted in bold, and the second is underlined. Color code: vermillion for $< 0.5$, orange for $[0.5, 0.75)$, yellow for $[0.75, 0.85)$, and bluish-green for $\geq 0.85$.

different training setups by varying $N$ across CSBM, $\alpha$-CSBM, and DLightSB-M. For the same set of methods, we experiment with two loss functions: KL and MSE.

In the main text, we report only the conditional metrics, as they more accurately reflect the performance of the EOT/SB solvers. In Appendix D.2 we provide experiments to validate conditional metrics against the unconditional ones. Further experimental details are provided in Appendix C.

**High-Dimensional Gaussian Mixtures.** Here, we report results on the high-dimensional Gaussian mixture benchmark constructed as in §3.3 using the methods from §4. Visual results are shown in Figure 1 for $q^{\text{gauss}}$ ($\gamma = 0.02$) and $q^{\text{unif}}$ ($\gamma = 0.005$), while additional plots provided in Appendix D.2. The qualitative results are presented in Tables 1a, 1b, 3, 4, 5 and 6.

First, across all benchmarks, each baseline underperforms for reasons specific to its construction. In particular, *Reference* consistently degrades, indicating that the scalar value function $v^*$ in (9) strongly reweights the reference process and yields a challenging $q^*$ to approximate. The *Independent* baseline deteriorates when the stochasticity parameter is small, while *Feature-wise SB* degrades as the dimension increases. Together, these failures highlight the target properties that our benchmarks are designed to test in EOT/SB solvers.

Second, DLightSB consistently achieves the strongest performance across all setups. We attribute this to the DLightSB solver being constructed according to the same principle underlying our benchmark. DLightSB-M, which incorporates the same inductive bias, attains comparable results with a slight drop in metrics, likely due to the additional variance introduced by the KL minimization loss. While this alignment could be seen as a limitation of the benchmark, we take the opposite view: DLightSB(-M) can be treated as an oracle-like method in this setting, since its inductive bias makes it less informative as a measure of general solver performance. For an analysis of the reverse benchmark designed to probe this bias, see Appendix D.1.

Finally, CSBM and $\alpha$-CSBM perform noticeably worse than DLightSB(-M). Notably, $\alpha$-CSBM attains comparable quality to CSBM while halving the computational cost, making it a more efficient alternative. Increasing $N$ generally improves the metrics. As for the loss function, KL consistently outperforms MSE, likely because MSE minimizes pointwise squared error and yields over-smoothed solutions that blur modes. This effect can be observed in Figure 2.

## 6 DISCUSSION

Our work addresses key gaps in discrete-space EOT/SB research by introducing the first standardized benchmark for these methods, along with new approaches. This contribution provides ground-truth data and consistent evaluation metrics for rigorous assessment of underlying solvers. The benchmark reveals fundamental limitations of both existing and proposed approaches: DLightSB(-M) faces severe memory constraints in high dimensions, while CSBM and $\alpha$-CSBM remain sensitive to parameter choices and require long training times. These findings underscore the need for more scalable architectures and stable training procedures, and guide future research in this area.

**Reproducibility.** We provide the experimental details in Appendix C and the code required to reproduce the conducted experiments is available in this repository (see `README.md` for details).

**LLM Usage.** Large Language Models (LLMs) were used only to assist with rephrasing sentences and improving the clarity of the text. All scientific content, results, and interpretations in this paper were developed solely by the authors.

## ACKNOWLEDGMENTS

The work was supported by the grant for research centers in the field of AI provided by the Ministry of Economic Development of the Russian Federation in accordance with the agreement 000000C313925P4F0002 and the agreement №139-10-2025-033.

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

## A  PROOFS

*Proof of Theorem 3.1.* We start from the expression of the static SB problem in (3)

$$\min_{q\in\Pi(p_0,p_1)} \text{KL}\left(q(x_0,x_1)\|q^{\text{ref}}(x_0,x_1)\right) =$$

$$= \min_{q\in\Pi(p_0,p_1)} -H(q) - \sum_{x_0,x_1} q(x_0,x_1)\log q^{\text{ref}}(x_1|x_0) - \text{const}$$

$$= \min_{q\in\Pi(p_0,p_1)} \sum_{x_0,x_1} q(x_0,x_1)\log q(x_0,x_1) - \sum_{x_0,x_1} q(x_0,x_1)\log q^{\text{ref}}(x_1|x_0) - \text{const} \tag{20}$$

Noting that the joint distribution can be written as $q(x_0,x_1) = q(x_0)q(x_1|x_0) = p_0(x_0)q(x_1|x_0)$, and enforcing the marginal constraints $\sum_{x_0} p_0(x_0)q(x_1|x_0) = p_1(x_1)$ and $\sum_{x_1} q(x_1|x_0) = 1$ (equivalently $q(x_0) = p_0(x_0)$), the corresponding Lagrangian can be formulated as

$$\mathcal{C}(q) = \sum_{x_0,x_1} p_0(x_0)q(x_1|x_0)\log\left(p_0(x_0)q(x_1|x_0)\right) - \sum_{x_0,x_1} p_0(x_0)q(x_1|x_0)\log q^{\text{ref}}(x_1|x_0) +$$

$$+ \sum_{x_1} \lambda(x_1)\left(\sum_{x_0} q(x_1|x_0)p_0(x_0) - p_1(x_1)\right) + \sum_{x_0} \tau(x_0)\left(\sum_{x_1} q(x_1|x_0) - p_0(x_0)\right)$$

$$= \underbrace{\sum_{x_0,x_1} p_0(x_0)q(x_1|x_0)\log p_0(x_0)}_{=\sum_{x_0} p_0(x_0)\log p_0(x_0)} + \sum_{x_0,x_1} p_0(x_0)q(x_1|x_0)\log q(x_1|x_0) - \tag{21}$$

$$- \sum_{x_0,x_1} p_0(x_0)q(x_1|x_0)\log q^{\text{ref}}(x_1|x_0) + \sum_{x_1} \lambda(x_1)\left(\sum_{x_0} q(x_1|x_0)p_0(x_0) - p_1(x_1)\right)$$

$$+ \sum_{x_0} \tau(x_0)\left(\sum_{x_1} q(x_1|x_0) - 1\right)$$

where $\lambda(x_1)$ and $\tau(x_0)$ denote the Lagrange multipliers associated with the marginal constraints on $x_1$ and $x_0$, respectively. Taking the pointwise partial derivative of $\mathcal{C}(q)$ with respect to $q(x_1|x_0)$

$$\frac{\partial\mathcal{C}}{\partial q(x_1|x_0)} = p_0(x_0)\left(\log q(x_1|x_0) + 1\right) - p_0(x_0)\log q^{\text{ref}}(x_1|x_0) + \lambda(x_1)p_0(x_0) + \tau(x_0) = 0 \tag{22}$$

Therefore, the process $q$ can be written as

$$q(x_1|x_0) = \exp(-\lambda(x_1) - 1)q^{\text{ref}}(x_1|x_0)\exp\left(-\frac{\tau(x_0)}{p_0(x_0)}\right) \tag{23}$$

We can then define

$$v(x_1) := \exp(-1 - \lambda(x_1)), \qquad \psi(x_0) := \exp\left(-\frac{\tau(x_0)}{p_0(x_0)}\right).$$

Then $q(x_1|x_0) = \psi(x_0)q^{\text{ref}}(x_1|x_0)v(x_1)$ and the normalization constraint $\sum_{x_1} q(x_1|x_0) = 1$ forces

$$\psi(x_0) = \frac{1}{\sum_{x_1} v(x_1)q^{\text{ref}}(x_1|x_0)} =: \frac{1}{c(x_0)}.$$

Thus the optimal conditional distribution $q^*(x_1|x_0)$ is uniquely defined by

$$q^*(x_1|x_0) = \frac{1}{c^*(x_0)} v^*(x_1)q^{\text{ref}}(x_1|x_0), \qquad \text{with } c^*(x_0) = \sum_{x_1} v^*(x_1)q^{\text{ref}}(x_1|x_0). \tag{24}$$

By construction $q^*(x_0) = p_0(x_0)$, and $p_1(x_1) := q^*(x_1) = \sum_{x_0} p_0(x_0)q^*(x_1|x_0)$ is its second marginal, so $q^* \in \Pi(p_0, p_1)$. Because the KL divergence is strictly convex and the feasible set $\Pi(p_0, p_1)$ is convex, the first-order conditions are sufficient for optimality. Hence $q^*$ is the unique minimizer of $\min_{q \in \Pi(p_0,p_1)} \mathrm{KL}\left(q\|q^{\mathrm{ref}}\right)$. Therefore $q^*$ in Equation (24) together with the reference process $q^{\mathrm{ref}}$ defines the discrete-space EOT/SB between $p_0$ and $p_1$.

$\square$

*Proof of Proposition 3.1.* Assuming the CP parameterization introduced in (8), and further assuming that the reference process factorizes across dimensions as $q^{\mathrm{ref}}(x_1|x_0) = \prod_{d=1}^{D} q^{\mathrm{ref}}(x_1^d|x_0^d)$, the normalized conditional distribution $q^*(x_1|x_0)$ in (7) can be rewritten as

$$
\begin{aligned}
q^*(x_1|x_0) &= \frac{1}{c^*(x_0)} \left( \sum_{k=1}^{K} \beta_k \prod_{d=1}^{D} r_k^d[x_1^d] \right) \prod_{d=1}^{D} q^{\mathrm{ref}}(x_1^d|x_0^d) \\
&= \frac{1}{c^*(x_0)} \sum_{k=1}^{K} \beta_k \prod_{d=1}^{D} r_k^d[x_1^d]\, q^{\mathrm{ref}}(x_1^d|x_0^d),
\end{aligned}
\tag{25}
$$

where the reference factors can be merged with the rank-1 components because they are independent of the mixture index $k$ and factorize over dimensions. From here, it is possible to obtain the normalizing constant $c^*(x_0)$ by summing over all possible values of $x_1 \in \mathcal{X} = \mathbb{S}^D$, where $x_1^d \in \{0, \ldots, S-1\}$. The normalizing constant can then be rewritten as

$$
\begin{aligned}
c^*(x_0) &= \sum_{x_1 \in \mathbb{S}^D} \sum_{k=1}^{K} \beta_k \prod_{d=1}^{D} r_k^d[x_1^d]\, q^{\mathrm{ref}}(x_1^d|x_0^d) \\
&= \sum_{k=1}^{K} \beta_k \sum_{x_1 \in \mathbb{S}^D} \prod_{d=1}^{D} r_k^d[x_1^d]\, q^{\mathrm{ref}}(x_1^d|x_0^d) \\
&= \sum_{k=1}^{K} \beta_k \prod_{d=1}^{D} \sum_{x_1^d=0}^{S-1} r_k^d[x_1^d]\, q^{\mathrm{ref}}(x_1^d|x_0^d),
\end{aligned}
\tag{26}
$$

where $\sum_{x_1 \in \mathbb{S}^D} = \sum_{x_1^1=0}^{S-1} \sum_{x_1^2=0}^{S-1} \cdots \sum_{x_1^D=0}^{S-1}$. The exchange between the product and the sum is valid here because the summation is separable across dimensions, i.e., each factor depends only on its corresponding coordinate $x_1^d$. $\square$

*Proof of Proposition 4.1.* We start from the standard KL minimization problem from the LightSB paper (Korotin et al., 2024) and define it in discrete space.

$$
\begin{aligned}
\mathrm{KL}\left(q^*\|q\right) &= \sum_{x_0,x_1} q^*(x_0,x_1) \log\left( \frac{q^*(x_0,x_1)}{q(x_0,x_1)} \right) = \sum_{x_0,x_1} q^* \log q^*(x_0,x_1) - \sum_{x_0,x_1} q^* \log q(x_0,x_1) = \\
&= -H(q^*) - \sum_{x_0,x_1} q^*(x_0,x_1) \log q(x_0,x_1) = -H(q^*) - \sum_{x_0,x_1} q^*(x_0,x_1) \log\left(q(x_0)q(x_1|x_0)\right) \\
&= -H(q^*) - \sum_{x_0,x_1} q^*(x_0,x_1) \log \underbrace{q(x_0)}_{=p_0(x_0)} - \sum_{x_0,x_1} q^*(x_0,x_1) \log q(x_1|x_0) = \\
&= -H(q^*) - \sum_{x_0} \log p_0(x_0) \underbrace{\sum_{x_1} q^*(x_0,x_1)}_{=q^*(x_0)=p_0(x_0)} - \sum_{x_0,x_1} q^*(x_0,x_1) \log q(x_1|x_0)
\end{aligned}
$$

Now using (7) on $q(x_1|x_0)$ we can get

$$\mathrm{KL}\left(q^*\|q\right) = -H(q^*) - \sum_{x_0} \log p_0(x_0)p_0(x_0) - \sum_{x_0,x_1} q^*(x_0,x_1) \log \left(\frac{v^*(x_1)}{c^*(x_0)} q^{\mathrm{ref}}(x_1|x_0)\right) =$$

$$= \underbrace{-H(q^*) - \sum_{x_0} \log p_0(x_0)p_0(x_0) - \sum_{x_0,x_1} q^*(x_0,x_1) \log q^{\mathrm{ref}}(x_1|x_0)}_{=-\mathcal{L}^*} -$$

$$- \sum_{x_0,x_1} q^*(x_0,x_1) \log \left(\frac{v^*(x_1)}{c^*(x_0)}\right) =$$

$$= -\mathcal{L}^* + \sum_{x_0,x_1} q^*(x_0,x_1) \log c^*(x_0) - \sum_{x_0,x_1} q^*(x_0,x_1) \log v^*(x_1) =$$

$$= \sum_{x_0} p_0^*(x_0) \log c^*(x_0) - \sum_{x_1} q^*(x_1) \log v^*(x_1) - \mathcal{L}^*$$

$$= \mathbb{E}_{p_0(x_0)}\left[\log c_\theta(x_0)\right] - \mathbb{E}_{p_1(x_1)}\left[\log v_\theta(x_1)\right] - \mathcal{L}^*,$$

That concludes the proof. $\qquad\square$

*Proof of Proposition 4.2.*

$$\mathrm{KL}\left(r(x_0,x_{\mathrm{in}},x_1)\|q^{\mathrm{SB}}(x_0,x_{\mathrm{in}},x_1)\right) =$$

$$= \mathrm{KL}\left(r(x_0,x_1)\|q^{\mathrm{SB}}(x_0,x_1)\right) + \underbrace{\mathrm{KL}\left(q^{\mathrm{ref}}(x_{\mathrm{in}}|x_0,x_1)\|q^{\mathrm{ref}}(x_{\mathrm{in}}|x_0,x_1)\right)}_{=0} = \quad (27)$$

$$= \underbrace{\sum_{x_0,x_1} r(x_0,x_1) \log r(x_0,x_1)}_{=-H(r(x_0,x_1))} - \sum_{x_0,x_1} r(x_0,x_1) \log q^{\mathrm{SB}}(x_0,x_1) =$$

$$= -H(r(x_0,x_1)) - \sum_{x_0,x_1} r(x_0,x_1) \log \frac{v^{\mathrm{SB}}(x_1)q^{\mathrm{ref}}(x_1|x_0)}{c^{\mathrm{SB}}(x_0)} = \quad (28)$$

$$= -H(r(x_0,x_1)) - \sum_{x_0,x_1} r(x_0,x_1) \log v^{\mathrm{SB}}(x_1) -$$

$$- \sum_{x_0,x_1} r(x_0,x_1) \log q^{\mathrm{ref}}(x_1|x_0) + \sum_{x_0,x_1} r(x_0,x_1) \log c^{\mathrm{SB}}(x_0) =$$

$$= -H(r(x_0,x_1)) - \sum_{x_1} \log v^{\mathrm{SB}}(x_1) \underbrace{r(x_1)}_{=p(x_1)=q^*(x_1)} \underbrace{\sum_{x_0} r(x_0|x_1)}_{=1=\sum_{x_0} q^*(x_0|x_1)} - \quad (29)$$

$$- \sum_{x_0,x_1} r(x_0,x_1) \log q^{\mathrm{ref}}(x_1|x_0) + \sum_{x_0} \log c^{\mathrm{SB}}(x_0) \underbrace{r(x_0)}_{=p(x_0)=q^*(x_0)} \underbrace{\sum_{x_1} r(x_1|x_0)}_{=1=\sum_{x_1} q^*(x_1|x_0)} = \quad (30)$$

$$= \underbrace{-H(r(x_0,x_1)) - \sum_{x_0,x_1} r(x_0,x_1) \log q^{\mathrm{ref}}(x_1|x_0)}_{=C_1} - \sum_{x_0,x_1} q^*(x_0,x_1) \log \frac{v^{\mathrm{SB}}(x_1)}{c^{\mathrm{SB}}(x_0)} =$$

$$= C_1 - \sum_{x_0,x_1} q^*(x_0,x_1) \log \frac{v^{\mathrm{SB}}(x_1)}{c^{\mathrm{SB}}(x_0)} -$$

$$\underbrace{- \sum_{x_0,x_1} q^*(x_0,x_1) \log q^{\mathrm{ref}}(x_1|x_0) + \sum_{x_0,x_1} q^*(x_0,x_1) \log q^{\mathrm{ref}}(x_1|x_0)}_{=0} = \quad (31)$$

$$= - \sum_{x_0, x_1} q^*(x_0, x_1) \log \frac{v^{\text{SB}}(x_1) q^{\text{ref}}(x_1|x_0)}{c^{\text{SB}}(x_0)} + C_1 + \underbrace{\sum_{x_0, x_1} q^*(x_0, x_1) \log q^{\text{ref}}(x_1|x_0)}_{=C_2} =$$

$$= C_2 - \sum_{x_0, x_1} q^*(x_0, x_1) \log q^{\text{SB}}(x_0, x_1) +$$

$$+ \underbrace{\sum_{x_0, x_1} q^*(x_0, x_1) \log q^*(x_0, x_1) - \sum_{x_0, x_1} q^*(x_0, x_1) \log q^*(x_0, x_1)}_{=0} = \quad (32)$$

$$= \sum_{x_0, x_1} q^*(x_0, x_1) \log \frac{q^*(x_0, x_1)}{q^{\text{SB}}(x_0, x_1)} + \underbrace{C_2 - \sum_{x_0, x_1} q^*(x_0, x_1) \log q^*(x_0, x_1)}_{C_3} =$$

$$= \text{KL}\left(q^*(x_0, x_1) \| q^{\text{SB}}(x_0, x_1)\right) + C_3$$

In (27), we use the disintegration of the KL divergence to transition from the dynamic to the static formulation. In (28), we apply our parameterization from (7). Next, in (29) and (30), we use the properties of the reciprocal process $r$, which has the true marginals at $t = 0$ and $t = 1$. In (31), we add a zero term to introduce $q^{\text{ref}}(x_1|x_0)$ with the expectation taken over the optimal joint distribution $q^*(x_0, x_1)$. Finally, in (32), we obtain the entropy term, completing the expression for the desired KL divergence. This establishes that optimizing $\text{KL}\left(r(x_0, x_1) \| q^{\text{SB}}(x_0, x_1)\right)$ with respect to $q^{\text{SB}}$ is equivalent to optimizing $\text{KL}\left(q^*(x_0, x_1) \| q^{\text{SB}}(x_0, x_1)\right)$ up to an additive constant. $\qquad \square$

*Proof of Proposition 3.2.* We first derive the transitional distributions of the SB by recalling its well-known characterization (Georgiou & Pavon, 2015, Thm. 2):

$$q^*\left(x_{t_n}|x_{t_{n-1}}\right) = q^{\text{ref}}\left(x_{t_n}|x_{t_{n-1}}\right) \frac{\phi_{t_n}^*(x_{t_n})}{\phi_{t_{n-1}}^*(x_{t_{n-1}})}, \qquad \phi_{t_n}^*(x_{t_n}) = \mathbb{E}_{q^{\text{ref}}(x_1|x_{t_n})}\left[v^*(x_1)\right].$$

Using the CP parametrization of $v^*$ from (8) and exploiting the factorization of $q^{\text{ref}}$, the scalar-valued functions $\phi_{t_n}$ can be written as:

$$\phi_{t_n}^*(x_{t_n}) = \sum_{k=1}^{K} \beta_k \prod_{d=1}^{D} \mathbb{E}_{q^{\text{ref}}(x_1^d|x_{t_n}^d)}\left[r_k^d[x_1^d]\right] = \sum_{k=1}^{K} \beta_k \prod_{d=1}^{D} \underbrace{\sum_{x_1^d=0}^{S-1} q^{\text{ref}}(x_1^d|x_{t_n}^d) \, r_k^d[x_1^d]}_{u_{k,t_n}^d[x_{t_n}^d]},$$

where $u_{k,t_n}^d$ satisfy the following recursive relation:

$$u_{k,t_n}^d[x_{t_n}^d] = \sum_{x_{t_{n+1}}^d=0}^{S-1} q^{\text{ref}}(x_{t_{n+1}}^d|x_{t_n}^d) \, u_{k,t_{n+1}}^d[x_{t_{n+1}}^d], \qquad u_{k,1}^d = r_k^d.$$

Thus, we obtain the following transition distributions:

$$q^*(x_{t_n}|x_{t_{n-1}}) \propto q^{\text{ref}}(x_{t_n}|x_{t_{n-1}}) \sum_{k=1}^{K} \beta_k \prod_{j=1}^{D} u_{k,t_n}^j[x_{t_n}^j]. \qquad (33)$$

To obtain the $d$-th marginal transition distribution, we marginalize over $x_{t_n}^{-d} := \{x_{t_n}^j\}_{j \neq d}$ as follows:

$$q^*\left(x_{t_n}^d|x_{t_{n-1}}\right) \propto \sum_{x_{t_n}^{-d}} \left(\prod_{j=1}^{D} q^{\text{ref}}(x_{t_n}^j|x_{t_{n-1}}^j)\right)\left(\sum_{k=1}^{K} \beta_k \prod_{j=1}^{D} u_{k,t_n}^j[x_{t_n}^j]\right) =$$

$$= q^{\text{ref}}(x_{t_n}^d|x_{t_{n-1}}^d) \sum_{k=1}^{K} \beta_k u_{k,t_n}^d[x_{t_n}^d] \prod_{\substack{j=1 \\ j \neq d}}^{D} \underbrace{\sum_{x_{t_n}^j} q^{\text{ref}}(x_{t_n}^j|x_{t_{n-1}}^j) \, u_{k,t_n}^j[x_{t_n}^j]}_{u_{k,t_{n-1}}^j[x_{t_{n-1}}^j] \text{ (by recursion)}}.$$

Finally, we obtain the desired expression up to normalization:

$$q^* \left( x_{t_n}^d | x_{t_{n-1}} \right) \propto [Q^{\text{ref}}]_{x_{t_{n-1}}^d, x_{t_n}^d} \sum_{k=1}^{K} \beta_k u_{k,t_n}^d [x_{t_n}^d] \prod_{\substack{j=1 \\ j \neq d}}^{D} u_{k,t_{n-1}}^j [x_{t_{n-1}}^j]. \tag{34}$$

$\square$

## B  CLOSED-FORM OF THE CONDITIONAL DISTRIBUTION FOR THE UNIFORM REFERENCE PROCESS

It is also useful to derive a closed-form expression for $q^{\text{ref}}(x_1^d | x_0^d)$. Since this conditional distribution is given by the $(N+1)$-step transition of the reference chain, it can be obtained from the $(N+1)$-th power of the transition matrix $Q^{\text{ref}}$. In the uniform case, this yields:

$$\overline{Q}_{N+1}^{\text{ref}} = \left( 1 - \gamma \frac{S}{S-1} \right)^{N+1} \mathbb{I} + \frac{1 - \left( 1 - \gamma \frac{S}{S-1} \right)^{N+1}}{S} \mathbf{1}\mathbf{1}^\top, \tag{35}$$

where $\mathbf{1} = [1, \ldots, 1]^\top \in \mathbb{R}^S$ is a vector of ones and $\mathbb{I}$ is an $S \times S$ identity matrix.

*Proof of Equation 35.* Let $Q$ be the transition matrix in (4), rewritten as

$$Q = (1 - \gamma)I + \frac{\gamma}{S-1}(\mathbf{1}\mathbf{1}^\top - I)$$

$$= \left( 1 - \gamma \frac{S}{S-1} \right) I + \frac{\gamma}{S-1} \mathbf{1}\mathbf{1}^\top,$$

where $I$ is the identity matrix and $\mathbf{1}\mathbf{1}^\top$ is the all-ones matrix. Let

$$a = 1 - \gamma \frac{S}{S-1}, \quad b = \frac{\gamma}{S-1},$$

so that $Q = aI + b\mathbf{1}\mathbf{1}^\top$ and note that $a + Sb = 1$. We compute $Q^{N+1}$ using the binomial expansion. Since $I$ and $\mathbf{1}\mathbf{1}^\top$ commute:

$$Q^n = (aI + b\mathbf{1}\mathbf{1}^\top)^n$$

$$= \sum_{k=0}^{n} \binom{n}{k} a^{n-k} b^k (\mathbf{1}\mathbf{1}^\top)^k.$$

Using $(\mathbf{1}\mathbf{1}^\top)^k = S^{k-1}\mathbf{1}\mathbf{1}^\top$ for $k \geq 1$ and separating the $k = 0$ term:

$$Q^n = a^n I + \sum_{k=1}^{n} \binom{n}{k} a^{n-k} b^k S^{k-1} \mathbf{1}\mathbf{1}^\top$$

$$= a^n I + \frac{1}{S} \left( \sum_{k=1}^{n} \binom{n}{k} a^{n-k} (bS)^k \right) \mathbf{1}\mathbf{1}^\top.$$

The binomial expansion gives:

$$(a + bS)^n = \sum_{k=0}^{n} \binom{n}{k} a^{n-k} (bS)^k = a^n + \sum_{k=1}^{n} \binom{n}{k} a^{n-k} (bS)^k.$$

Since $a + bS = 1$, we have $(a + bS)^n = 1$, so $\sum_{k=1}^{n} \binom{n}{k} a^{n-k} (bS)^k = 1 - a^n$. Thus,

$$Q^n = a^n I + \frac{1 - a^n}{S} \mathbf{1}\mathbf{1}^\top.$$

Substituting $n = N+1$ and $a = 1 - \gamma\frac{S}{S-1}$ yields

$$q^{\text{ref}}(x_1^d|x_0^d) = Q^{N+1} = \left(1 - \gamma\frac{S}{S-1}\right)^{N+1} I + \frac{1 - \left(1 - \gamma\frac{S}{S-1}\right)^{N+1}}{S}\mathbf{1}\mathbf{1}^\top.$$

$\square$

From here it can be seen that $\overline{Q}_{N+1}^{\text{ref}}$ converges to $(1/S)\mathbf{1}\mathbf{1}^\top$ when $(N+1) \to \infty$, that is a uniform distribution over the number of categories $S$. In the case of the Gaussian reference process, the closed-form expression can also be obtained, but it is much more complex.

## C  EXPERIMENT DETAILS

This section provides detailed descriptions of all methods and their configurations.

**Shared Aspects.** Across all experiments, we use the AdamW optimizer with fixed `beta` values of $0.95$ and $0.99$. For the high-dimensional Gaussian benchmark (§5.2). Notably, for diffusion-based methods, we fully sample the Markov chain, in contrast to Austin et al. (2021), which applies an `argmax` operation at the final timestep.

**CSBM and $\alpha$-CSBM.** For CSBM and $\alpha$-CSBM, we use the official implementation from Ksenofontov & Korotin (2025):

$$\text{https://github.com/gregkseno/csbm.}$$

To stabilize training and improve final performance, we apply Exponential Moving Average (EMA) parameter updates with a decay rate of $0.999$, tuned consistently across all experiments. Unlike Austin et al. (2021), we omit the $L_{\text{simple}}$ loss during training. We employ a simple MLP with three hidden layers of size $[128, 128, 128]$ and ReLU activations. Time conditioning is implemented via an embedding layer of the same size as dimensions, $D$. Both methods are trained for $5$ D-IMF iterations, using $120\,000$ gradient updates in the first iteration and $40\,000$ in each subsequent iteration. For $\alpha$-CSBM, we use a learning rate of $10^{-3}$ and halve the batch size for training a single model, following De Bortoli et al. (2024). For CSBM, we use a learning rate of $10^{-4}$. The D-IMF procedure for both solvers is initialized using an independent joint distribution $(q^0(x_0, x_1) = p_0(x_0)p_1(x_1))$.

**DLightSB and DLightSB-M.** For all benchmark experiments, both methods use $K = 1000$ components initialized from data samples and are trained for $100\,000$ gradient updates. The learning rate is set to $10^{-2}$ for both, with DLightSB-M using independent joint distribution $(q^0(x_0, x_1) = p_0(x_0)p_1(x_1))$.

**Computational Resources and Training Time.** All high-dimensional Gaussian mixture benchmark experiments were conducted on 1 A100 GPU unless otherwise specified, with training times reported inclusive of evaluation. For $D = 2$, training is relatively short: CSBM and $\alpha$-CSBM each complete within about $5$ hours, DLightSB-M within $4$ hours, and DLightSB in roughly $20$ minutes. For $D = 64$, CSBM completes in under $14$ hours, $\alpha$-CSBM in under $9$ hours, DLightSB-M in just under $2$ days (on 2 A100 GPUs), and DLightSB in under $7$ hours.

## D  ADDITIONAL EXPERIMENTS

### D.1  REVERSE BENCHMARK

In this section, we try to overcome the inherited inductive bias of DLightSB(-M) solvers. By construction, the forward conditional distribution $q^*(x_1|x_0)$ admits a CP decomposition, while the reverse distribution $q^*(x_0|x_1)$ does not. As a result, when the benchmark is used in the reverse direction with the same marginals $p_0$ and $p_1$, DLightSB(-M) methods can no longer rely on the inductive bias that benefits them in the forward setup.

Unfortunately, in this setup, the true conditional distributions are not available, so we cannot compute conditional metrics. To overcome this restriction, we decided to compute the Classifier Two Sample Test (Lopez-Paz & Oquab, 2017, C2ST) metric, ROC AUC of classifier between pairs $(x_0, x_1) \sim p_1(x_1)q^*(x_0|x_1)$ and $(\hat{x}_0, x_1) \sim p_1(x_1)q_\theta(x_0|x_1)$. As the classifier, we used two layer MLP with ReLU activations that takes as input the concatenation of one-hot vectors of $x_0$ and $x_1$. We present C2ST scores in the following table.

| Method | Loss | $N+1$ | $D=2$ gaussian 0.02 | 0.05 | uniform 0.005 | 0.01 | $D=16$ gaussian 0.02 | 0.05 | uniform 0.005 | 0.01 | $D=64$ gaussian 0.02 | 0.05 | uniform 0.005 | 0.01 |
|---|---|---|---|---|---|---|---|---|---|---|---|---|---|---|
| DLightSB | – | – | 0.93 | 1.00 | 1.00 | 0.99 | 0.96 | 0.97 | 0.99 | 1.00 | 0.97 | 0.99 | 0.99 | 0.99 |
| CSBM | KL | 16 | 0.99 | 0.99 | 1.00 | 1.00 | 0.98 | 0.99 | 1.00 | 0.99 | 0.99 | 0.99 | 0.99 | 1.00 |
| | | 64 | 1.00 | 1.00 | 0.99 | 1.00 | 0.99 | 0.98 | 0.99 | 0.98 | 1.00 | 1.00 | 0.99 | 1.00 |
| | MSE | 16 | 0.95 | 1.00 | 0.99 | 1.00 | 1.00 | 0.98 | 1.00 | 0.99 | 0.99 | 1.00 | 0.98 | 1.00 |
| | | 64 | 0.90 | 0.99 | 0.99 | 0.98 | 0.99 | 0.99 | 1.00 | 0.97 | 0.99 | 1.00 | 1.00 | 1.00 |

Table 2: C2ST metric ($\uparrow$) on our high-dimensional Gaussian mixture benchmark. Color code: vermillion for $< 0.5$, orange for $[0.5, 0.75)$, yellow for $[0.75, 0.85)$, and bluish-green for $\geq 0.85$.

As can be seen from Table 2, computed metric values are not informative. Across all methods, the metric values are nearly identical, indicating that the classifier is not capable of distinguishing generated samples from real ones. As a result, we decided to discard this metric.

## D.2 ADDITIONAL METRICS AND PLOTS

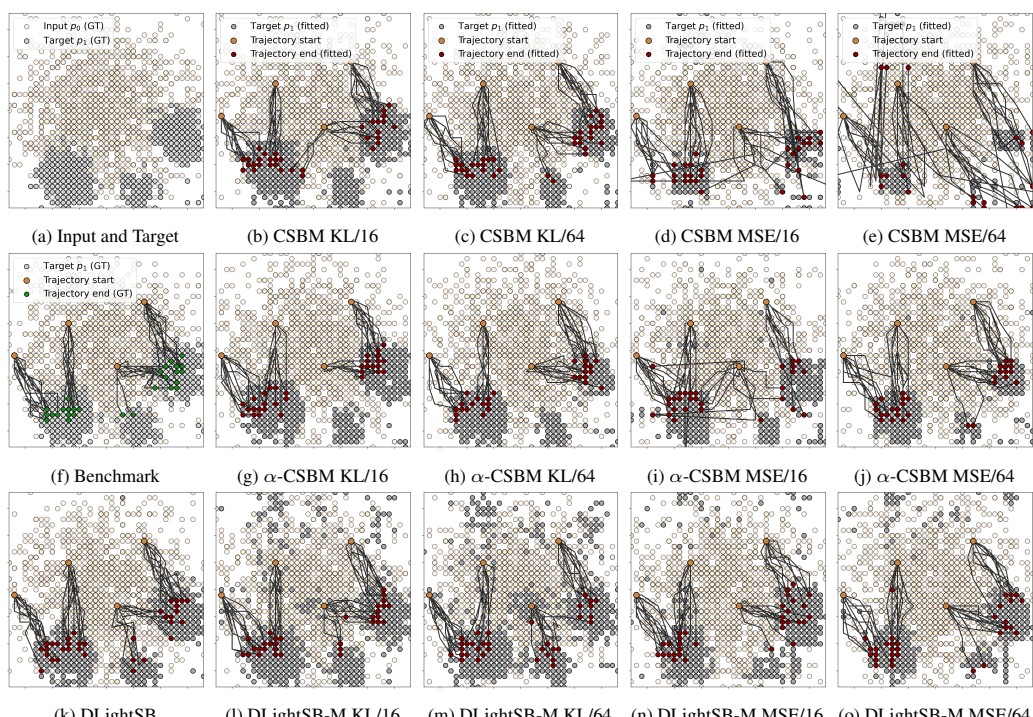

(a) Input and Target  (b) CSBM KL/16  (c) CSBM KL/64  (d) CSBM MSE/16  (e) CSBM MSE/64

(f) Benchmark  (g) $\alpha$-CSBM KL/16  (h) $\alpha$-CSBM KL/64  (i) $\alpha$-CSBM MSE/16  (j) $\alpha$-CSBM MSE/64

(k) DLightSB  (l) DLightSB-M KL/16  (m) DLightSB-M KL/64  (n) DLightSB-M MSE/16  (o) DLightSB-M MSE/64

Figure 2: Samples from all methods on the high-dimensional Gaussian mixture benchmark using the Gaussian reference process $q^{\text{gauss}}$ with $\gamma = 0.02$.

| Method | Loss | N+1 | D=2 gaussian 0.02 | 0.05 | uniform 0.005 | 0.01 | D=16 gaussian 0.02 | 0.05 | uniform 0.005 | 0.01 | D=64 gaussian 0.02 | 0.05 | uniform 0.005 | 0.01 |
|---|---|---|---|---|---|---|---|---|---|---|---|---|---|---|
| *Reference* | – | – | 0.5 | 0.6 | 0.3 | 0.5 | 0.5 | 0.7 | 0.3 | 0.5 | 0.5 | 0.6 | 0.3 | 0.5 |
| *Feature-wise SB* | – | – | 0.0 | 0.0 | 0.0 | 0.0 | **0.0** | **0.0** | 0.1 | 0.1 | 0.0 | 0.0 | 0.0 | 0.0 |
| DLightSB | – | – | **0.0** | **0.0** | **0.0** | **0.0** | 1.8 | **0.0** | **0.0** | **0.0** | 0.3 | 0.1 | 0.1 | 0.1 |
| CSBM | KL | 16 | 8.0 | 32.1 | 4.7 | 8.1 | 18.4 | 189.1 | 70.8 | 104.9 | 16.8 | 22.0 | 15.2 | 14.1 |
| | | 64 | 1.5 | 7.4 | 1.0 | 1.5 | 2.8 | 21.0 | 7.9 | 8.8 | 22.8 | 2.3 | 4.0 | 3.6 |
| | MSE | 16 | 14.0 | 23.8 | 7.9 | 7.4 | 28.7 | 182.9 | 108.0 | 156.8 | 47.5 | 24.9 | 36.8 | 25.8 |
| | | 64 | 9.5 | 10.3 | 3.2 | 3.1 | 6.7 | 45.0 | 25.2 | 32.1 | 59.3 | 2.1 | 14.8 | 10.9 |
| $\alpha$-CSBM | KL | 16 | 6.4 | 23.7 | 4.2 | 6.4 | 13.3 | 103.4 | 61.1 | 81.5 | 26.1 | 3.5 | 4.9 | 5.2 |
| | | 64 | 1.3 | 4.9 | 0.7 | 0.7 | 3.5 | 1.1 | 2.1 | 2.2 | 24.5 | 2.1 | 4.5 | 4.6 |
| | MSE | 16 | 7.6 | 17.5 | 5.6 | 5.2 | 16.1 | 112.3 | 52.8 | 72.3 | 46.2 | 3.4 | 17.9 | 12.0 |
| | | 64 | 1.0 | 3.1 | 0.9 | 0.8 | 9.4 | 4.7 | 5.6 | 4.4 | 58.8 | 2.0 | 15.5 | 11.0 |
| DLightSB-M | KL | 16 | 0.1 | **0.0** | **0.0** | **0.0** | 3.2 | 0.1 | 0.2 | 0.2 | 6.2 | 1.3 | 26.4 | 22.3 |
| | | 64 | 0.2 | **0.0** | **0.0** | **0.0** | 1.2 | 0.2 | 0.4 | 0.3 | 6.6 | 0.8 | 22.9 | 7.3 |
| | MSE | 16 | 0.8 | **0.0** | 0.2 | **0.0** | 22.0 | 0.2 | 0.5 | 0.3 | 26.3 | 1.4 | 8.8 | 48.3 |
| | | 64 | 1.0 | **0.0** | 0.2 | 0.1 | 24.8 | 0.3 | 0.6 | 0.4 | 58.8 | 2.3 | 41.0 | 17.9 |

Table 3: Trajectory KL divergence (↓) on the high-dimensional Gaussian mixture benchmark. The best-performing method is highlighted in bold, and the second is underlined. Color code: bluish-green for $< 0.5$, yellow for $[0.5, 2)$, orange for $[2, 10)$, and vermillion for $\geq 10$.

| Method | Loss | N+1 | D=2 gaussian 0.02 | 0.05 | uniform 0.005 | 0.01 | D=16 gaussian 0.02 | 0.05 | uniform 0.005 | 0.01 | D=64 gaussian 0.02 | 0.05 | uniform 0.005 | 0.01 |
|---|---|---|---|---|---|---|---|---|---|---|---|---|---|---|
| *Reference* | – | – | 2.1 | 3.0 | 0.8 | 1.3 | 2.2 | 4.6 | 1.4 | 1.5 | 1.4 | 3.5 | 1.0 | 1.2 |
| *Feature-wise SB* | – | – | 0.2 | 0.2 | **0.0** | **0.0** | 0.2 | 0.5 | 0.3 | 0.3 | 0.0 | 0.4 | 0.3 | 0.4 |
| DLightSB | – | – | **0.0** | **0.0** | **0.0** | **0.0** | 5.7 | **0.0** | **0.0** | **0.0** | 0.3 | 0.1 | 0.1 | 0.1 |
| CSBM | KL | 16 | 1.5 | 3.7 | 1.0 | 1.1 | 7.7 | 22.1 | 50.9 | 71.4 | 15.8 | 13.0 | 16.6 | 20.2 |
| | | 64 | 0.4 | 1.8 | 0.4 | 0.4 | 2.7 | 7.1 | 10.3 | 10.8 | 22.3 | 2.5 | 11.3 | 10.8 |
| | MSE | 16 | 4.2 | 2.5 | 3.3 | 4.3 | 8.1 | 19.5 | 88.3 | 122.3 | 55.4 | 12.2 | 41.8 | 34.7 |
| | | 64 | 10.2 | 1.8 | 2.1 | 2.2 | 4.4 | 9.1 | 22.2 | 24.4 | 75.6 | 2.6 | 38.5 | 39.0 |
| $\alpha$-CSBM | KL | 16 | 1.7 | 3.3 | 0.8 | 0.9 | 8.1 | 17.5 | 41.8 | 53.8 | 23.6 | 5.0 | 20.5 | 27.3 |
| | | 64 | 0.7 | 1.4 | 0.4 | 0.3 | 4.4 | 1.0 | 7.3 | 10.7 | 17.1 | 3.0 | 17.1 | 16.2 |
| | MSE | 16 | 2.7 | 3.0 | 1.7 | 1.1 | 9.6 | 18.5 | 52.5 | 75.4 | 54.1 | 5.6 | 83.6 | 89.5 |
| | | 64 | 0.5 | 1.1 | 1.4 | 1.3 | 11.4 | 2.8 | 16.9 | 20.0 | 75.9 | 3.3 | 46.7 | 48.3 |
| DLightSB-M | KL | 16 | 0.5 | **0.0** | 0.1 | 0.2 | 21.7 | 1.0 | 7.4 | 5.9 | 6.4 | 2.3 | 322.2 | 329.7 |
| | | 64 | 0.7 | 0.1 | 0.3 | 0.4 | 8.2 | 1.4 | 12.3 | 11.7 | 6.8 | 1.9 | 292.6 | 110.7 |
| | MSE | 16 | 1.1 | 0.3 | 0.3 | 0.2 | 182.3 | 4.5 | 6.5 | 6.6 | 25.4 | 15.8 | 70.3 | 405.0 |
| | | 64 | 1.4 | 0.5 | 0.4 | 0.2 | 156.4 | 8.2 | 8.4 | 8.9 | 52.4 | 27.6 | 313.8 | 104.1 |

Table 4: Trajectory reverse KL divergence (↓) on the high-dimensional Gaussian mixture benchmark. The best-performing method is highlighted in bold, and the second is underlined. Color code: bluish-green for $< 0.5$, yellow for $[0.5, 2)$, orange for $[2, 10)$, and vermillion for $\geq 10$.

| | | | D=2 | | | | D=16 | | | | D=64 | | | |
|---|---|---|---|---|---|---|---|---|---|---|---|---|---|---|
| | | | gaussian | | uniform | | gaussian | | uniform | | gaussian | | uniform | |
| Method | Loss | N+1 | 0.02 | 0.05 | 0.005 | 0.01 | 0.02 | 0.05 | 0.005 | 0.01 | 0.02 | 0.05 | 0.005 | 0.01 |
| *Independent* | – | – | **0.98** | **0.99** | **0.98** | **0.98** | **0.99** | **0.99** | **0.98** | **0.98** | **0.99** | **0.98** | **0.98** | **0.98** |
| *Reference* | – | – | 0.32 | 0.48 | 0.41 | 0.43 | 0.41 | 0.34 | 0.41 | 0.42 | 0.56 | 0.49 | 0.48 | 0.47 |
| *Feature-wise SB* | – | – | 0.92 | **0.99** | 0.98 | 0.98 | 0.92 | 0.98 | **0.98** | **0.98** | 0.98 | **0.98** | **0.98** | **0.98** |
| DLightSB | – | – | 0.97 | 0.97 | **0.98** | **0.98** | 0.89 | 0.97 | 0.96 | **0.98** | 0.97 | 0.95 | 0.97 | 0.97 |
| CSBM | KL | 16 | 0.77 | 0.72 | 0.92 | 0.91 | 0.82 | 0.78 | 0.85 | 0.79 | 0.91 | 0.90 | 0.84 | 0.89 |
| | | 64 | 0.91 | 0.89 | 0.96 | 0.97 | 0.91 | 0.85 | 0.92 | 0.93 | 0.94 | 0.94 | 0.79 | 0.88 |
| | MSE | 16 | 0.52 | 0.72 | 0.84 | 0.86 | 0.82 | 0.74 | 0.80 | 0.74 | 0.81 | 0.89 | 0.81 | 0.81 |
| | | 64 | 0.37 | 0.84 | 0.84 | 0.81 | 0.88 | 0.83 | 0.84 | 0.90 | 0.74 | 0.96 | 0.83 | 0.82 |
| α-CSBM | KL | 16 | 0.74 | 0.75 | 0.93 | 0.91 | 0.79 | 0.81 | 0.85 | 0.78 | 0.93 | 0.89 | 0.81 | 0.87 |
| | | 64 | 0.82 | 0.89 | 0.96 | 0.97 | 0.92 | 0.93 | 0.93 | 0.95 | 0.95 | 0.93 | 0.80 | 0.86 |
| | MSE | 16 | 0.69 | 0.73 | 0.89 | 0.90 | 0.79 | 0.80 | 0.85 | 0.83 | 0.82 | 0.93 | 0.83 | 0.82 |
| | | 64 | 0.92 | 0.88 | 0.89 | 0.93 | 0.80 | 0.90 | 0.87 | 0.90 | 0.74 | 0.93 | 0.82 | 0.85 |
| DLightSB-M | KL | 16 | 0.90 | 0.96 | 0.97 | 0.97 | 0.86 | 0.93 | 0.96 | 0.96 | 0.89 | 0.80 | 0.83 | 0.86 |
| | | 64 | 0.88 | 0.95 | 0.95 | 0.96 | 0.90 | 0.95 | 0.95 | 0.96 | 0.89 | 0.90 | 0.80 | 0.93 |
| | MSE | 16 | 0.75 | 0.95 | 0.85 | 0.93 | 0.69 | 0.96 | 0.93 | 0.92 | 0.72 | 0.88 | 0.89 | 0.79 |
| | | 64 | 0.73 | 0.95 | 0.81 | 0.92 | 0.65 | 0.95 | 0.94 | 0.89 | 0.62 | 0.89 | 0.68 | 0.83 |

Table 5: Shape Score metric (↑) on our high-dimensional Gaussian mixture benchmark. The best-performing method is highlighted in bold, and the second is underlined. Color code: vermillion for $< 0.5$, orange for $[0.5, 0.75)$, yellow for $[0.75, 0.85)$, and bluish-green for $\geq 0.85$.

| | | | D=2 | | | | D=16 | | | | D=64 | | | |
|---|---|---|---|---|---|---|---|---|---|---|---|---|---|---|
| | | | gaussian | | uniform | | gaussian | | uniform | | gaussian | | uniform | |
| Method | Loss | N+1 | 0.02 | 0.05 | 0.005 | 0.01 | 0.02 | 0.05 | 0.005 | 0.01 | 0.02 | 0.05 | 0.005 | 0.01 |
| *Independent* | – | – | **0.97** | **0.97** | **0.96** | **0.96** | **0.96** | **0.96** | **0.94** | **0.95** | **0.95** | **0.92** | **0.92** | **0.92** |
| *Reference* | – | – | 0.17 | 0.30 | 0.25 | 0.29 | 0.19 | 0.12 | 0.14 | 0.14 | 0.37 | 0.23 | 0.23 | 0.20 |
| *Feature-wise SB* | – | – | 0.71 | 0.70 | 0.78 | 0.82 | 0.81 | 0.66 | 0.55 | 0.51 | 0.93 | 0.67 | 0.73 | 0.68 |
| DLightSB | – | 16 | 0.96 | 0.96 | **0.96** | **0.96** | 0.84 | 0.95 | 0.93 | 0.94 | 0.94 | 0.90 | 0.91 | 0.91 |
| CSBM | KL | 16 | 0.70 | 0.64 | 0.87 | 0.87 | 0.73 | 0.66 | 0.75 | 0.67 | 0.85 | 0.83 | 0.78 | 0.83 |
| | | 64 | 0.89 | 0.85 | 0.93 | 0.93 | 0.87 | 0.81 | 0.87 | 0.88 | 0.90 | 0.89 | 0.73 | 0.83 |
| | MSE | 16 | 0.50 | 0.65 | 0.77 | 0.79 | 0.73 | 0.62 | 0.69 | 0.62 | 0.72 | 0.81 | 0.74 | 0.76 |
| | | 64 | 0.32 | 0.78 | 0.76 | 0.74 | 0.81 | 0.77 | 0.78 | 0.83 | 0.63 | 0.90 | 0.77 | 0.77 |
| α-CSBM | KL | 16 | 0.66 | 0.66 | 0.89 | 0.87 | 0.71 | 0.71 | 0.78 | 0.70 | 0.88 | 0.83 | 0.74 | 0.81 |
| | | 64 | 0.81 | 0.85 | 0.93 | 0.93 | 0.89 | 0.90 | 0.88 | 0.91 | 0.91 | 0.88 | 0.74 | 0.81 |
| | MSE | 16 | 0.63 | 0.64 | 0.83 | 0.83 | 0.71 | 0.69 | 0.76 | 0.73 | 0.73 | 0.87 | 0.75 | 0.75 |
| | | 64 | 0.89 | 0.84 | 0.83 | 0.86 | 0.72 | 0.87 | 0.81 | 0.84 | 0.63 | 0.88 | 0.75 | 0.79 |
| DLightSB-M | KL | 16 | 0.87 | 0.95 | 0.94 | 0.95 | 0.78 | 0.91 | 0.91 | 0.92 | 0.83 | 0.74 | 0.64 | 0.67 |
| | | 64 | 0.85 | 0.93 | 0.93 | 0.94 | 0.84 | 0.92 | 0.91 | 0.92 | 0.83 | 0.85 | 0.64 | 0.83 |
| | MSE | 16 | 0.67 | 0.93 | 0.80 | 0.91 | 0.54 | 0.92 | 0.88 | 0.88 | 0.60 | 0.83 | 0.79 | 0.51 |
| | | 64 | 0.65 | 0.92 | 0.76 | 0.89 | 0.49 | 0.91 | 0.88 | 0.85 | 0.46 | 0.82 | 0.48 | 0.72 |

Table 6: Trend Score (↑) on our high-dimensional Gaussian mixture benchmark. The best-performing method is highlighted in bold, and the second is underlined. Color code: vermillion for $< 0.5$, orange for $[0.5, 0.75)$, yellow for $[0.75, 0.85)$, and bluish-green for $\geq 0.85$.

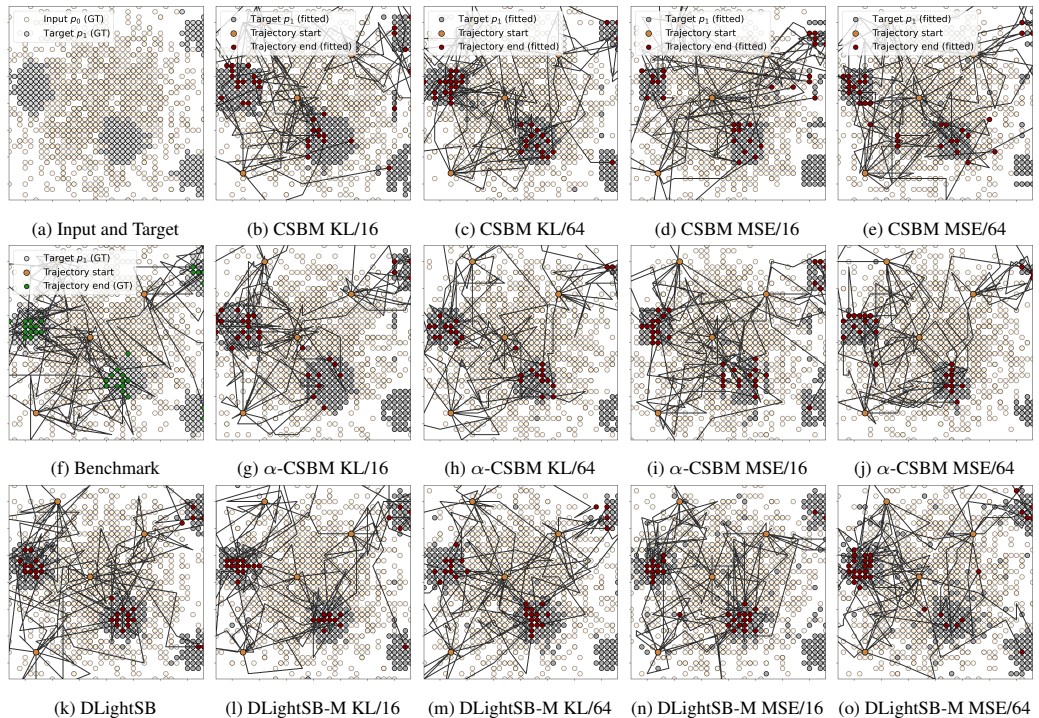

Figure 3: Samples from all methods on the high-dimensional Gaussian mixture benchmark using the Gaussian reference process $q^{\text{gauss}}$ with $\gamma = 0.05$.

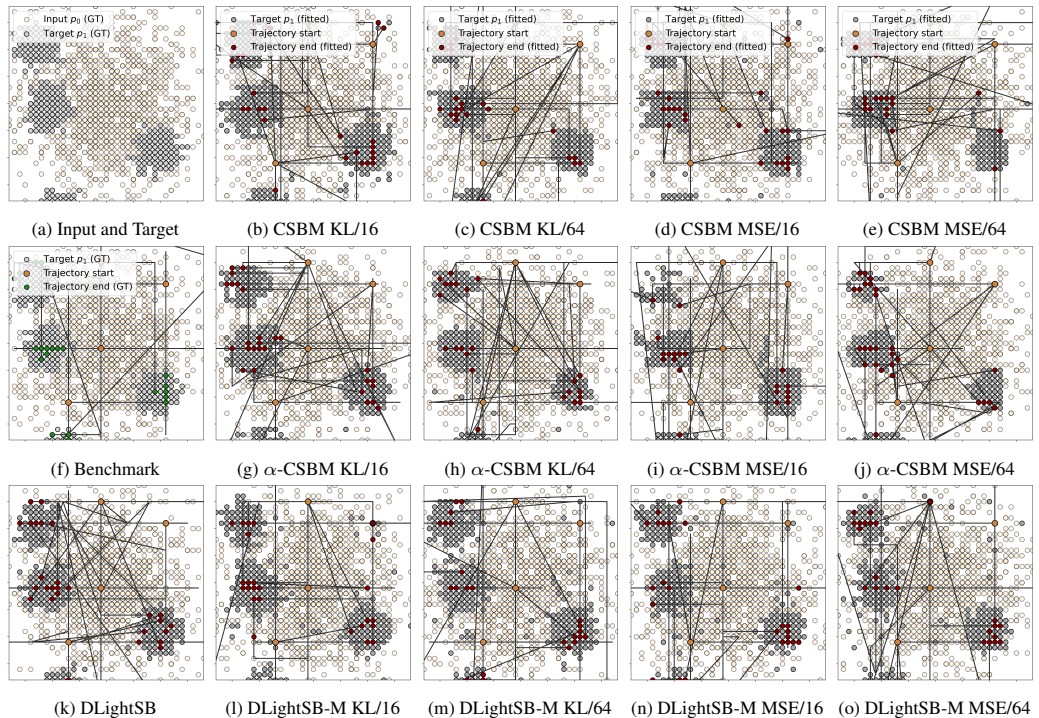

Figure 4: Samples from all methods on the high-dimensional Gaussian mixture benchmark using the uniform reference process $q^{\text{unif}}$ with $\gamma = 0.005$.

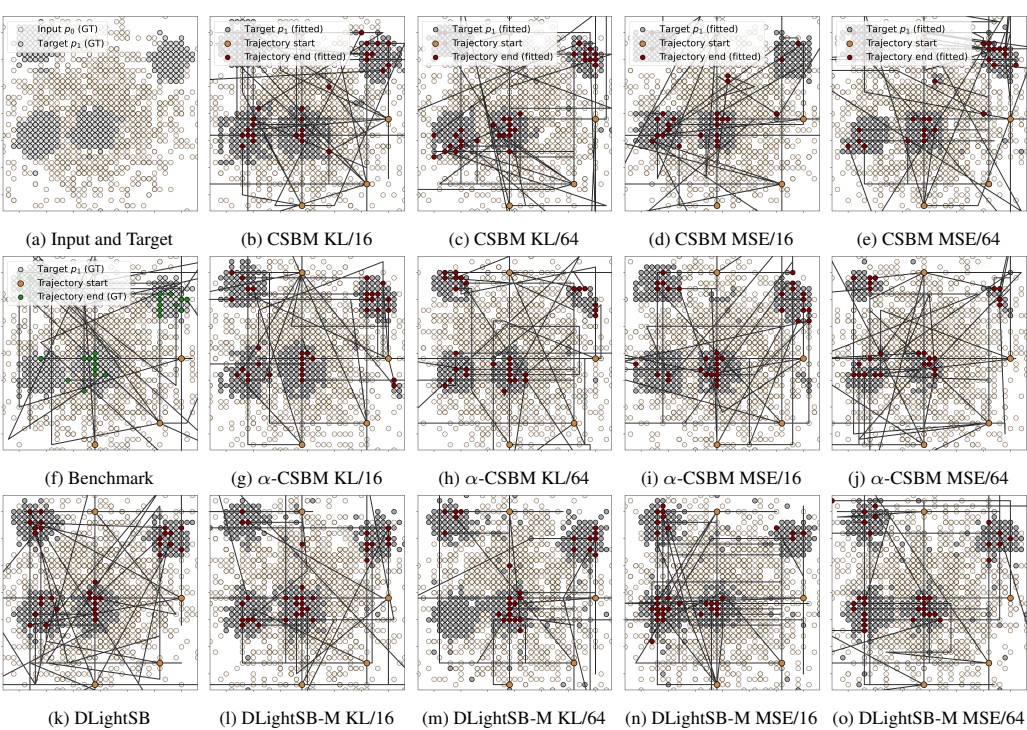

Figure 5: Samples from all methods on the high-dimensional Gaussian mixture benchmark using the uniform reference process $q^{\text{unif}}$ with $\gamma = 0.01$.

