# OpenReview forum: "Entering the Era of Discrete Diffusion Models: A Benchmark for Schrödinger Bridges and Entropic Optimal Transport"
_ICLR.cc/2026/Conference — ICLR 2026 Poster_

### Official Review · Reviewer_FM4X · 2025-10-26

**Soundness:** 3
**Presentation:** 2
**Contribution:** 2
**Rating:** 4
**Confidence:** 4

**Summary:**

The authors proposed a mixture-like benchmark for SB on discrete spaces to evaluate the effectiveness of different methods. The authors also proposed a light version of DSB and a smoothing version of the CSBM algorithm, α-CSBM,  to enhance the training stability and efficiency. Interesting comparisons are made on toy examples to show the strength and weakness.

**Strengths:**

1. Providing a mixture example to evaluate the accuracy of different approaches is helpful to the community.

2. The light (low rank) and alpha smoothing do have some potential to accelerate the algorithm.

3. IMF is more appealing and interesting than the IPF style of algorithms.

**Weaknesses:**

1. Even the "expensive" version of the CSBM algorithm is evaluated on real-world image experiments. The light and alpha-smoothed algorithm is only evaluated on the mixture benchmark. This is far from sufficient.

2. The writing is not good. The current draft is a mixture of lots of components, like a baby mixture benchmark, 2 light algorithms (without sufficient discussions on theoretical properties or empirical evaluations). The scope is ambitious, but for each contribution itself, it is a bit minor.

3. Although low0

**Questions:**

1. Could you provide an additional section (like section 4.4 in CSBM) to verify the scalability?

2. i didn't get line 352: "be generalized to an arbitrary reference process qref, thereby enabling the application of the optimal projection in discrete space settings." If some algorithm accuracy is compromised (like low rank), does it still guarantee optimal projection? it is a bit confusing.

3. I haven't fully checked CSBM yet. Is the algorithm simulation-free or not? To me, low-rank may be interesting but the simulation-free property is more important for the scalability.

---

> ### Author Response · Authors · 2025-12-03
>
> We thank the reviewer for the review and comments. We will address the weaknesses and questions below.
>
> **[W1] and [Q1] Even the "expensive" version of the CSBM algorithm is evaluated on real-world image experiments. The light and alpha-smoothed algorithm is only evaluated on the mixture benchmark. This is far from sufficient. Could you provide an additional section (like section 4.4 in CSBM) to verify the scalability?**
>
> We believe there is a minor misunderstanding regarding the scope of our contributions. Our objective is not to evaluate how solvers scale across different tasks, but to provide a pipeline that allows SB methods on discrete spaces to be compared fairly. For this reason, we do not include additional scalability tests of the type presented in the related CSBM paper.
>
> **[W2] The writing is not good. The current draft is a mixture of lots of components, like a baby mixture benchmark, 2 light algorithms (without sufficient discussions on theoretical properties or empirical evaluations). The scope is ambitious, but for each contribution itself, it is a bit minor.**
>
> Regarding writing quality, we have revised several sections to improve clarity. All modifications in the revised version are highlighted in blue.
>
> Indeed, our work consists of several smaller contributions. However, each of them is necessary for constructing our main contribution, namely the benchmark itself. For example, introducing the DLightSB(-M) methods is essential for ensuring the integrity of the benchmark; see our response to [Weakness 3] for Reviewer MD4S. Finally, we view that the benchmark itself as an important step toward advancing research on SB solvers on discrete solver and therefore, we humbly believe that our overall contribution is not minor.
>
> **[W4] i didn't get line 352: "be generalized to an arbitrary reference process qref, thereby enabling the application of the optimal projection in discrete space settings." If some algorithm accuracy is compromised (like low rank), does it still guarantee optimal projection? it is a bit confusing.**
>
> If we understand the question correctly, it concerns whether, under a given parametrization (e.g., CP decomposition), Proposition 4.2 guarantees that the procedure yields the exact SB solution, i.e., $q^\* = q_\theta$. We believe the confusion arises from the use of the term "optimal". In our context, "optimal" simply indicates that the projection is defined directly onto the set of SB solutions (regardless of specific parametrization), rather than enforcing the reciprocal and Markovian properties iteratively as in the IMF procedure. Consequently, if a parametrization restricts (unlike CP decomposition, which is a universal approximator) the expressiveness of the model, the resulting solution may deviate from the exact SB, yet this does not contradict the proposition.
>
> **[W5] I haven't fully checked CSBM yet. Is the algorithm simulation-free or not? To me, low-rank may be interesting but the simulation-free property is more important for the scalability.**
>
> No, ($\alpha$-)CSBM methods are not simulation-free, as they require sampling after the first D-IMF iteration to build pairs for training. In contrast, DLightSB(-M) methods are indeed simulation-free. Thus, DLightSB(-M) inherits both desirable properties, which may be beneficial in certain applications.
>
> **Concluding remarks.** Kindly let us know if the clarifications provided address your concerns about our work. We're happy to discuss any remaining questions during the discussion phase. If you find our responses satisfactory, we would appreciate it if you could consider raising your score.

---

### Official Review · Reviewer_ECS7 · 2025-10-31

**Soundness:** 3
**Presentation:** 3
**Contribution:** 3
**Rating:** 6
**Confidence:** 3

**Summary:**

In this paper, an analytic benchmark for discrete optimal transport is developed. In order to assess capabilities of current algorithms for solving the schrodinger bridge problem on discrete state spaces, there is a need for a benchmark for which the ground truth solution is known. The authors construct this analytic solution by writing the optimal transition distribution as a scalar function multiplying the reference transition distribution. Then a specific mixture of factorized forms is proposed for the scalar function such that the true schrodinger bridge transition distribution becomes analytically calculable. The authors propose two new light schrodinger bridge algorithms for discrete state spaces and compare them with existing methods on their benchmark.

**Strengths:**

The use of the mixture of factorized functions for the scalar tilting function looks novel to me and is an interesting way to construct a non-trivial problem for which the schrodinger bridge is known. The same factorization technique is then also used to construct the Light Schrodinger algorithms which show very good performance compared to prior bridge matching approaches. It is important to have these Light Schrodinger bridge baseline algorithms for further comparison during algorithm development as they serve as useful reference points.

**Weaknesses:**

The benchmark seems to already be too easy because the DLightSB method is already almost saturating the performance metrics. This means the benchmark is already losing its discriminative power between the best methods. Ideally, the benchmark would contain sub-tasks for which all methods struggle. Could the task be made more difficult by increasing the number of mixture components in the scalar function?

In the introduction in line 47 you state that one motivation for introducing this benchmark is to check whether methods actually solve the SB problem separating out 'true algorithmic performance from artifacts of specific parameterizations'. However, then on L431 you hypothesize that DLightSB performs the best on this benchmark because it is built on the same factorization principle as the benchmark itself. This would then seem to go against the stated goal of the benchmark to try and not be influenced by specific parameterizations of the method. Perhaps the benchmark needs to be made more complex as to not be 'defeated' by the DLightSB algorithm.

The differences between your two proposed methods DLightSB and DLightSB-M could have been investigated more. You hypothesize that DLightSB-M works worse due to error accummulation. This could have been verified or falsified with a further experiment, for example, changing the number of discretization time steps in the diffusion process to see if the discrepancy between the two methods gets smaller.

Equations (19) and (20) are quite hard to parse due to the circular definition of the set \mathcal{S}. The set itself is used within its own definition in the argmax and makes the whole equation hard to understand.

Overall, I think the contribution of the rank-1 Canonical Polyadic decomposition for constructing both ground truth SBs and for learning LightSBs is a worthy contribution to the conference notwithstanding the limitations in evaluations of the proposed methods.

**Questions:**

Why do CSBM algorithms have better performance on the benchmark with higher dimensionality? I would expect that performance would degrade with higher dimensions given a fixed model capacity but perhaps because of the low dimensional structure in the benchmark this is not the case?

Wouldn't it be possible to create a metric that doesn't just take into account single marginals or pairs of marginals by looking at distribution metrics? For example you could take a given starting point x0, sample p(x1 | x0) from the learned SB and the true SB then compute a distributional metric between those two sample distributions. Then you could average these values over different draws of x0.

---

> ### Author Response · Authors · 2025-12-03
>
> We thank the reviewer for the review and comments. We will address the weaknesses and questions below.
>
> **[W1] The benchmark seems to already be too easy because the DLightSB method is already almost saturating the performance metrics. This means the benchmark is already losing its discriminative power between the best methods. Ideally, the benchmark would contain sub-tasks for which all methods struggle. Could the task be made more difficult by increasing the number of mixture components in the scalar function?**
>
> To address the concerns regarding the benchmark complexity, we kindly refer the reviewer to our response to [Weakness 2] for Reviewer MD4S, where we provide a full explanation.
>
> Regarding the suggestion to make the task more difficult by increasing the number of mixture components, this is certainly possible. However, it does not substantially affect DLightSB(-M) performance due to its inductive bias, and therefore, we do not conduct such experiments.
>
> To properly address this limitation, we have investigated a **reverse benchmark** instead. Please refer to the discussion of the reverse benchmark in our response to question [Weakness 3] for Reviewer MD4S.
>
> **[W2] In line 47 you state that one motivation for introducing this benchmark is to check whether methods actually solve the SB problem separating out 'true algorithmic performance from artifacts of specific parameterizations'. However, then on L431 you hypothesize that DLightSB performs the best on this benchmark because it is built on the same factorization principle as the benchmark itself. This would then seem to go against the stated goal of the benchmark.**
>
> We respectfully disagree that this constitutes a contradiction. The lines 47 describe properties of a general benchmark. Our work aims to satisfy these properties, although DLightSB(-M) have parametrization choices that could influence their behavior on this benchmark. Nevertheless, we argue that this does not diminish the benchmark's value for assessing other solvers, as we already cover the main approaches within this inductive-bias class. For further details, we refer the reviewer to our response to [Weakness 3] for Reviewer MD4S.
>
> **[W3] The differences between your two proposed methods DLightSB and DLightSB-M could have been investigated more. You hypothesize that DLightSB-M works worse due to error accummulation. This could have been verified or falsified with a further experiment, for example, changing the number of discretization time steps in the diffusion process to see if the discrepancy between the two methods gets smaller.**
>
> We thank the reviewer for the suggestions. First, we kindly point out that our experiments currently include different number of time steps ($N+1 =$ {16, 64} for KL and MSE losses), they can be found in our results in Tables 1, 2, 4 and 5. Second, the primary focus of our paper lies in the benchmark itself. For this reason, we did not pursue more extensive ablation studies to further probe the capabilities of the solvers.
>
> **[W4] Equations (19) and (20) are quite hard to parse due to the circular definition of the set $\mathcal{S}$. The set itself is used within its own definition in the argmax and makes the whole equation hard to understand.**
>
> We apologize for the confusion, we have addressed it in the new revision.
>
> **[Q1] Why do CSBM algorithms have better performance on the benchmark with higher dimensionality? I would expect that performance would degrade with higher dimensions given a fixed model capacity but perhaps because of the low dimensional structure in the benchmark this is not the case?**
>
> This behavior arises because the shape and trend score metrics consider only the marginals and the joint distribution of element pairs $D$, respectively. This leads to an evaluation that is insensitive to the space dimensionality: even if the data lives in a higher-dimensional space, the metrics ignore any structure beyond pairwise correlations. The reported performance of CSBM algorithms slightly increases simply because the metrics fail to capture the true complexity of the underlying distributions. In this sense, the improved scores reflect a limitation of the metrics rather than a genuine gain in modeling ability.

---

> ### Author Response · Authors · 2025-12-03
>
> **[Q2] Wouldn't it be possible to create a metric that doesn't just take into account single marginals or pairs of marginals by looking at distribution metrics? For example you could take a given starting point x0, sample p(x1 | x0) from the learned SB and the true SB then compute a distributional metric between those two sample distributions. Then you could average these values over different draws of x0.**
>
> We thank the reviewer for this suggestion. The process described by the reviewer is what we actually call **conditional metrics**. As explained in lines 400-402 of the original manuscript, we sample multiple $x_1$ for every $x_0$ and then the resulting metrics are averaged. We report the conditional metrics in Tables 1 and 2 while the unconditional ones are in Tables 4 and 5 in Appendix D. The revised version of the paper includes a much clearer explanation in this regard in Lines 417-420
>
> **Concluding remarks.** Kindly let us know if the clarifications provided address your concerns about our work. We're happy to discuss any remaining questions during the discussion phase. If you find our responses satisfactory, we would appreciate it if you could consider raising your score.

---

### Official Review · Reviewer_dFbH · 2025-11-01

**Soundness:** 3
**Presentation:** 3
**Contribution:** 1
**Rating:** 4
**Confidence:** 4

**Summary:**

This paper presents a benchmark for SB methods on discrete spaces. The contributions of this work seem to be two-fold: (1) a theoretical investigation for constructing a benchmark for discrete SB problems and (2) an actual benchmark along with initial results, including three new implementations of SB algorithms. The authors constructed a practical benchmark based on Gaussian mixtures. The implemented DLightSB algorithms (discrete variants) achieved the best results.

**Strengths:**

Based on my reading, the paper has the following strengths.
* Solid theoretical investigation and a well-grounded evaluation strategy for discrete SB problems.
* An actual benchmark that will benefit the SB community.
* Comprehensive empirical experiments and new discrete SB algorithms

**Weaknesses:**

I believe this work in its current format lacks novelty in many regards.

* The theoretical arguments presented in this paper can be seen as either well-known or a reformulation of continuous SB work. I generally view expanding discrete theoretical work to continuous spaces as more challenging and practical, not the other way around.
* The proposed benchmark setup relies heavily on Gaussian mixtures, which were originally intended for representing distributions in continuous spaces, and this tarnishes one of the central contributions of the paper (focusing on the new discrete problem), as the statistics will share many characteristics with continuous problems.
* It is widely known in the SB community that LightSB algorithms excel in Gaussian mixture EOT benchmarks, since LightSB actually has a parametric prior of Gaussian mixture models (see Korotin et al. and Gushchin et al.). Therefore, I am somewhat certain that the authors were highly knowledgeable of this information before the experimental design and engineered DLightSB and DLightSB-M to show similarly good performance on their platform. In other words, these results contain very similar implications to the two LightSB papers, which makes the novelty of this paper weak, since not much new information is given beyond the references.
* The sheer number of benchmarks is lacking, and four important algorithms (DLightSB, DLightSB-M, and α-CSBM) can be generally considered as reformulations from the work of a single research group. The authors are encouraged to put some more effort into implementing other discrete formulations of competitive SB algorithms.

**Questions:**

* Is there a table that comprehensively summarizes the basic statistics of the GMM benchmarks? (D={2, 16, 64})

---

> ### Author Response · Authors · 2025-12-03
>
> We thank the reviewer for the review and comments. We will address the weaknesses and questions below.
>
> **[W.1] The theoretical arguments presented in this paper can be seen as either well-known or a reformulation of continuous SB work. I generally view expanding discrete theoretical work to continuous spaces as more challenging and practical, not the other way around.**
>
> We would like to emphasize that, while the theoretical components naturally parallel continuous SB work, establishing this framework is an important step for the discrete domain. This adaptation is necessary to enable the development of SB solvers for discrete data, which presents unique challenges not found in continuous spaces.
>
> **[W.2] The proposed benchmark setup relies heavily on Gaussian mixtures, which were originally intended for representing distributions in continuous spaces, and this tarnishes one of the central contributions of the paper (focusing on the new discrete problem), as the statistics will share many characteristics with continuous problems.**
>
> This is a very interesting observation, but we do not fully agree with it. It is important to note that the SB problem is defined not only by the marginals $p_0, p_1$ but also by the reference process $q^{ref}$. More importantly, it is the reference process that determines the underlying dynamics that the SB solution must follow. These dynamics are exactly what the benchmark evaluates when it compares $q^\*(x_1 | x_0)$ to $q_\theta(x_1 | x_0)$.
>
> Therefore, to capture discrete-space behavior, our benchmark includes a uniform reference process in addition to the Gaussian-like one. As shown in Figure 1, the resulting SB trajectories depend strongly on this choice. With the Gaussian-like reference, indeed the trajectories resemble those typically observed in continuous SB settings. However, when the uniform reference is used, the behavior changes significantly. The trajectories often jump between distant categories, which clearly reflects the dynamics only specific to discrete-state system.
>
> For this reason, including a uniform reference process in our opinion is sufficient to ensure that the benchmark captures the discrete statistical characteristics central to our study.
>
> **[W.3] ... LightSB algorithms excel in Gaussian mixture EOT benchmarks, since LightSB actually has a parametric prior of Gaussian mixture models. Therefore, I am somewhat certain that the authors were highly knowledgeable of this information before the experimental design and engineered DLightSB and DLightSB-M to show similarly good performance on their platform.**
>
> Thank you for this comment. We believe there may be a slight misunderstanding, as our goal is not to present DLightSB(-M) as state-of-the-art methods. As noted in the original manuscript (lines 466–467), their strong performance in the benchmark arises directly from their inductive bias (see also our response to [Weakness 3] for Reviewer MD4S on why this does not pose an issue). This should not be interpreted as an attempt to promote them as superior solvers. Instead, our primary goal is to design and construct a tractable benchmark for discrete SB methods based on CP decomposition, rather than to introduce entirely new solvers.
>
> **[W.4] The sheer number of benchmarks is lacking, and four important algorithms (DLightSB, DLightSB-M, and $\alpha$-CSBM) can be generally considered as reformulations from the work of a single research group. The authors are encouraged to put some more effort into implementing other discrete formulations of competitive SB algorithms.**
>
> We would like to clarify that our contribution does not primarily lie in implementing new solvers, but in designing and constructing a tractable benchmark for SB methods on discrete space. The limited number of methods we evaluate is not due to a lack of effort on our side; rather, the current literature offers very few SB solvers on discrete space, as this research direction is only beginning to emerge. We fully agree that developing additional solvers is important. This is precisely why we introduce the benchmark: it provides a straightforward and standardized evaluation path for future, entirely novel methods.
>
> **[Q.1] Is there a table that comprehensively summarizes the basic statistics of the GMM benchmarks? (D={2, 16, 64})**
>
> Unfortunately, it is not possible to provide this table as the means were generated randomly. We understand that this is an important point to keep the benchmark reproducible and we will update our code to save the generated means and standard deviations.
>
> **Concluding remarks.** Kindly let us know if the clarifications provided address your concerns about our work. We're happy to discuss any remaining questions during the discussion phase. If you find our responses satisfactory, we would appreciate it if you could consider raising your score.

---

### Official Review · Reviewer_MD5S · 2025-11-02

**Soundness:** 2
**Presentation:** 3
**Contribution:** 1
**Rating:** 2
**Confidence:** 4

**Summary:**

The paper identifies a valid and significant gap in the machine learning literature: the absence of a standardized, ground-truth benchmark for evaluating Schrödinger Bridge (SB) solvers on high-dimensional discrete state spaces. The authors' stated contributions are:
1.  A method for constructing benchmark pairs of probability distributions $(p_0, p_1)$ on discrete spaces ($\mathcal{X} = \mathbb{S}^D$) for which the analytical SB solution $q^*$ is known by design.
2.  The use of a Canonical Polyadic (CP) decomposition to parameterize a key scalar function ($v^*$), rendering the benchmark construction and sampling computationally tractable in high dimensions.
3.  The introduction of three new or adapted solvers (DLightSB, DLightSB-M, and $\alpha$-CSBM) as "byproducts" of this framework.
4.  An empirical evaluation of these new solvers against the existing CSBM method, using the proposed benchmark, which concludes that the DLightSB and DLightSB-M methods demonstrate superior performance.

Despite the commendable goal, the paper's execution suffers from several fundamental weaknesses in its methodology, evaluation, and framing that undermine its primary conclusions and limit its utility as a general benchmark for the community.

**Strengths:**

1. Given a source marginal, and a scalar valued function (what is known in the literature as potentials) it is easy to obtain a corresponding target marginal and the solution SB distribution. The construction is simple and practical and serves as a good synthetic test bed.

2. The parametrization of the scalar valued potential using rank-1 Canonical Polyadic decompositions seems to be versatile (in theory) given that they can act as universal approximators and possess nice theoretical properties.

**Weaknesses:**

### 1. Stated Theoretical Contribution: Benchmark Construction

The paper's central theoretical claim is the introduction of a method for constructing ground-truth benchmark pairs for the Schrödinger Bridge (SB) problem on high-dimensional discrete spaces. This method yields pairs of probability distributions $(p_0, p_1)$ for which the analytical SB solution $q^*$ is known by design, enabling rigorous evaluation.

This construction is formally presented in **Theorem 3.1**, which states:

1.  One begins with a known source (initial) distribution $p_0 \in \mathcal{P}(\mathcal{X})$.

2.  One also defines a scalar-valued function $v^*: \mathcal{X} \rightarrow \mathbb{R}$.

3.  The "ground-truth" joint distribution $q*$  is then constructed such that its $x_0$ marginal is $p_0$ and its conditional distribution $q ^* (x| y)$ is proportional to the product of this new function $v*$ and the reference transition kernel $q^{ref}(x_1|x_0)$:

    $$q ^* (x_1|x_0) \propto v^*(x_1) q^{ref}(x_1|x_0)$$,

or

   $$q ^* (x_1|x_0) =v^*(x_1) q^{ref}(x_1|x_0) / c(x_0).$$

4.  The target distribution $p_1$ is then *defined* as the second marginal of this $q ^* (x_0, x_1) = p_0(x_0) q ^* (x_1|x_0)$.
5.  The theorem concludes that the $q^*(x_0, x_1)$ constructed this way is the true static SB solution between the resulting $p_0$ and $p_1$.


### 2. Marginal Theoretical Novelty

The theoretical novelty of this construction (Theorem 3.1) is only marginal. It is a "reverse-engineered" benchmark based on a restatement of the well-established dual solution to the Entropic Optimal Transport (EOT) problem, which the paper itself shows is equivalent to the static SB problem (Section 2.3).

* **Standard EOT Duality:** The solution $\pi ^*$ to the EOT problem $\min_{\pi \in \Pi(p_0, p_1)} \text{KL}(\pi || R)$ is uniquely characterized by two potentials (Lagrange multipliers) $\phi$ and $\psi$ such that the optimal coupling has the form:

    $$\pi^*(x_0, x_1) \propto \phi(x_0) \psi(x_1) R(x_0, x_1)$$

* **Deriving Theorem 3.1 from first principles:** The paper's construction can be easily derived from the known result above.

    * The paper's *reference measure* $R(x_0, x_1)$ is $q^{ref}(x_0, x_1)$, which is assumed to have the $p_0$ marginal: $R(x_0, x_1) = p_0(x_0) q^{ref}(x_1|x_0)$.
    * Suppose we construct the optimal coupling as $q ^* (x_0, x_1) = \phi(x_0) v ^* (x_1) q^{ref}(x_0, x_1)$, and we want its first marginal to be $p_0$. Integrating out $x_1$ gives $p_0(x_0) = \phi(x_0) p_0(x_0) \sum_{x_1} v ^* (x_1) q^{ref}(x_1 | x_0)$, so canceling out $p_0$ terms gives $\phi(x_0) = 1/c(x_0) = 1/ \sum_{x_1} v ^* (x_1) q^{ref}(x_1 | x_0)$. This is exactly the paper's construction.

Since one can easily deduce the construction in Theorem 3.1 from a very well-known first principle in SB/EOT literature, it doesn't appear as a significant contribution in this work.

### 3. The "Single-Potential Problem" Simplification

The paper's methodology *chooses* to construct a benchmark for a simplified theoretical problem.

* The general, two-potential SB problem requires solving for *both* potentials $\phi(x_0)$ and $\psi(x_1)$ to satisfy the two marginal constraints $\pi_0 = p_0$ and $\pi_1 = p_1$.
* The paper's benchmark, the only unknown potential is for the output domain; the input domain's potential is then determined. (Note that the potential $\phi$ on the input space is determined by the potential $v ^*$ on the output domain and the reference transition kernel $q^{ref}(x_1 | x_0)$.)
* This theoretical simplification is crucial because the paper's "winning" solver, **DLightSB**, is *explicitly* a static, dual-problem solver designed to solve this very particular problem.



### 4. Theoretical Limitation of the Chosen Method

The paper's actual innovation is not Theorem 3.1, but the methodology to make it computationally tractable. The theoretical constructions above are trivial in low dimensions, but in the paper's high-dimensional setting ($\mathcal{X} = \mathbb{S}^D$), the potential $v ^*$ and the normalization constants are intractable, requiring sums over $S^D$ states.

The paper's *actual* contribution is the application of a Canonical Polyadic (CP) decomposition to parameterize $v^*$. As shown in **Proposition 3.1**, this factorization cleverly turns the intractable normalization sum into a product of sums, making the computation feasible. This is a practical, **engineering contribution**, not a new theory for constructing benchmarks.

This reliance on a CP parameterization for tractability imposes a severe *theoretical limitation* on the benchmark itself: it guarantees the ground-truth problem is trivially simple.

* The computational tractability of the CP method is only achieved if the rank $K$ is small.
* The authors, in their experiments, use **$K=4$** to generate the ground-truth data.
* This means the "challenging" benchmark problem is, by construction, an *extremely simple, low-rank function*.
* This defeats the entire purpose of a benchmark, which is to test a method's ability to solve a complex problem. The paper's theoretical framework *requires* the problem to be simple for the ground truth to be computable.


### 5. Fundamental Methodological Bias and "Apples-to-Oranges" Evaluation
The paper's central weakness is a circular and biased evaluation methodology. The primary conclusion—that DLightSB and DLightSB-M are the superior solvers—is an artifact of the benchmark's design rather than a demonstration of general algorithmic strength.

* **Architectural Circularity:** The tractability of the benchmark (Prop. 3.1) relies entirely on parameterizing the scalar function $v^*$ using a **CP decomposition**. The authors' "new" solvers, DLightSB and DLightSB-M, are *also* explicitly parameterized using the **exact same CP decomposition**. This creates an evaluation that is not a fair comparison of general SB solvers. Instead, it tests how well a CP-based model can fit a function already known to have a CP structure. The authors explicitly acknowledge this circularity: "We attribute this to the benchmark pairs being built on the same principle used by the DLightSB solver".

* **"Apples-to-Oranges" Comparison:** The benchmark problem itself represents a *simplified* version of the full SB problem. The authors' "generative" construction (Theorem 3.1) fixes $p_0$ and the reference dynamics, which effectively reduces the problem to solving for a *single* static potential, $v^*$. The paper then compares two fundamentally different classes of algorithms:
    * **DLightSB (The "Apple"):** A **static, dual-problem solver** whose entire purpose is to fit this single potential $v_\theta$.
    * **CSBM (The "Orange"):** A **dynamic, primal-problem solver** based on Iterative Markovian Fitting (D-IMF), which is designed to learn the *full time-dependent transition probabilities* $m(x_{t_n} | x_{t_{n-1}})$.

This evaluation is inherently unfair. A specialized static solver (DLightSB) that is custom-built for this simplified problem (and given the correct architecture) is predictably going to outperform a general-purpose dynamic solver (CSBM) that is designed for a much harder task.

### 6. Triviality of the Benchmark Task
The reliance on the CP parameterization for tractability forces the ground-truth benchmark to be exceptionally simple, defeating the purpose of a benchmark, which should test a challenging problem.

* The ground-truth benchmark is generated using a CP decomposition with a rank of **$K=4$**.
* The authors' "winning" solvers, DLightSB and DLightSB-M, are then initialized with a capacity of **$K=1000$** components.

The task is thus to fit a simple $K=4$ mixture model using a massively over-parameterized ($K=1000$) model that shares the *identical* functional form. The 250-fold capacity advantage, combined with the correct inductive bias, makes the problem trivial for the DLightSB solvers. A benchmark that is this simple and architecturally-biased provides little insight into how these solvers would perform on complex, real-world problems.

### 7. Flawed Evaluation Metrics and Exclusion of Relevant Competitors
The paper's evaluation protocol is indirect and avoids the most rigorous tests of its own contributions by excluding the most relevant class of competing algorithms.

* **Exclusion of Neural EOT Solvers:** The paper's "Remark" dismisses all "tabular EOT" solvers as "non-generative." This is a strawman argument that ignores the modern, active field of research that *does* learn generalizable, generative couplings by parameterizing the dual potentials ($\phi, \psi$) with neural networks. A fair and informative benchmark would have *required* a comparison between **DLightSB (CP-potential)** and a **Neural EOT Solver (MLP-potential)**. This "apples-to-apples" comparison would have directly tested the paper's only true innovation: the claim that a CP decomposition is a good inductive bias for this problem.

* **Inadequate "Black-Box" Metrics:** By excluding other potential-fitting methods, the paper is forced into the "apples-to-oranges" (DLightSB vs. CSBM) comparison. Because these models learn different objects (potentials vs. dynamics), a direct comparison is impossible. The authors are thus forced to use weak, indirect metrics: **Shape Score** and **Trend Score**. These metrics are inadequate for two reasons:
    1.  **They Don't Measure the *Real* Ground Truth:** The *entire point* of the benchmark is that the true potential $v*$ is known. The most rigorous metric would be a direct comparison of the learned potential $v_\theta$ to $v*$. The paper avoids this.

    2.  **They Don't Guarantee Optimality:** These metrics only measure low-order statistics of the *output samples*. They confirm that the model can approximate the *final marginal* $p_1$, but they provide no guarantee that the model has learned the *true, optimal transport coupling* $q ^*$.

**Questions:**

#### **On the Benchmark Design and Evaluation Bias**

1.  The paper's primary "winning" solver, DLightSB, is based on a Canonical Polyadic (CP) decomposition. The benchmark's ground truth is *also* constructed using a CP decomposition (Prop 3.1).  How can we be sure that the superior results of DLightSB are not an artifact of it being given the correct inductive bias, a bias that competing methods like CSBM (using an MLP) do not have?

2.  The paper's results show that DLightSB "wins" on a benchmark built on "the same principle used by the DLightSB solver." Does this not confirm a circular evaluation? How would DLightSB perform on a benchmark constructed with a *different* non-factorizable structure, such as one based on a deep neural potential?

3.  The ground-truth benchmark is generated with a low rank ($K=4$), while the DLightSB solver is given a capacity of $K=1000$. Does this 250-fold capacity advantage not make the task trivial for a solver that already shares the problem's functional form? What insights can be gained from a benchmark that is this simple and architecturally-matched to the solver?

#### **On the Choice of Methods and Comparisons**

4.  The remark in Section 2.4 dismisses "tabular EOT" solvers as non-generative. This seems to overlook the entire class of modern EOT solvers that parameterize the dual potentials ($\phi, \psi$) with neural networks, which are both generative and generalizable. Why were these potential-fitting neural EOT methods—which would provide a true "apples-to-apples" comparison against DLightSB—not included as a baseline?

5.  The evaluation compares DLightSB (a *static, dual-potential solver*) against CSBM (a *dynamic, primal-path solver*). Is this not a "apples-to-oranges" comparison? DLightSB solves a simpler, static problem, while CSBM solves a much harder, dynamic one. How can we conclude DLightSB is a better "Schrödinger Bridge solver" from this comparison?

#### **On the Choice of Evaluation Metrics**

6.  The entire purpose of the benchmark is that the ground-truth potential $v^*$ is known. Why, then, were the evaluation metrics (Shape Score, Trend Score) based on comparing *output samples*? Why did the authors not use the most direct and rigorous metric: the error between the learned potential $v_\theta$ and the true potential $v*$?

#### **On the Theoretical Contributions and Framing**

7.  Theorem 3.1 appears to be a restatement of the well-known dual solution to the EOT problem, where one potential is set to 1 by construction. Can the authors clarify the theoretical novelty of this theorem beyond this restatement?

8.  The paper's main innovation appears to be the *engineering* choice of using a CP decomposition to make the benchmark tractable. However, this choice seems to *require* the ground-truth problem to be low-rank ($K=4$). Does this not create a fundamental trade-off where the benchmark can be either (a) computable but trivial, or (b) complex but not computable?

10. The paper's "winning" solvers (DLightSB, DLightSB-M) are admitted to have "severe memory constraints" and become "prohibitive for high-dimensional data." This seems to create a contradictory message. What is the value of a solver that only "wins" on a simple, biased benchmark and is not scalable to the very high-dimensional problems it claims to address?

---

> ### Author Response · Authors · 2025-12-03
>
> We thank the reviewer for the review and comments. We will address the weaknesses and questions below. To make the discussion more concise and easier to follow, we grouped several related weaknesses and questions. We use [Weakness 1] to refer to these grouped answers and [W.1] to refer to the reviewer's comments/questions.
>
> **[Weakness 1] ([W.1], [W.2] and [Q.7]) Theoretical Contribution: Benchmark Construction. Theorem 3.1 appears to be a restatement of the well-known dual solution to the EOT problem, where one potential is set to 1 by construction. Can the authors clarify the theoretical novelty of this theorem beyond this restatement?**
>
> We would like to emphasize that, even if the theoretical component may appear "marginal", this theorem is essential for constructing the benchmark. Regardless of whether one follows your proof approach or ours, the benchmark cannot be built without it. We apologize if our manuscript gave the impression that this theorem is our core contribution. As you correctly pointed out, our main contribution is proposing a **computationally tractable** (based on CP-decomposition) benchmark for discrete data. We make this distinction explicit in the contribution list of the revised manuscript.
>
> **[Weakness 2] ([W.3], [W.5-2], [W.6], [Q.1], [Q.2], [Q.3] and [Q.9]) Fundamental Methodological Bias and "Apples-to-Oranges" Evaluation. Cyclic evaluation. Overparameterized solver vs. underparameterized benchmark. Value of the DLightSB(-M) solvers. Scalability of the DLightSB(-M) solvers.**
>
> Indeed, we agree that the DLightSB solvers have inductive bias. However, we do not claim that these solvers are state-of-the-art. Moreover, we argue that this limitation does not pose a problem for evaluating future solvers on our benchmark. We have made sufficient effort to ensure that all methods that inherit the inductive bias from the benchmark construction are already implemented in our paper. To further clarify this, in the discussion section of the revised version, we explicitly highlight this point to prevent future methods from exploiting the inductive bias in a way that could compromise the integrity of the benchmark.
>
> Another important point is that retaining DLightSB methods is still valuable, even in a biased setup. They serve as approximate "oracle" methods that confirm the appropriateness of our benchmark and metrics: the oracle methods achieve scores close to $1$, while the baselines discussed in [Weakness 1] obtain very low values.
>
> We saw the problem and have already tried to overcome this limitation by proposing the following experiment.
>
> **Reverse benchmark.** By construction, the forward conditional distribution $q^\*(x_1 | x_0)$ admits a CP decomposition, while the reverse distribution $q^*(x_0 | x_1)$ does not. As a result, when the benchmark is used in the reverse direction with the same marginals $p_0$ and $p_1$, DLightSB(-M) methods can no longer rely on the inductive bias that benefits them in the forward setup.
>
> Unfortunately, in this setup, the true conditional distributions are not available, so we cannot compute conditional metrics. To overcome this restriction, we decided to compute the Classifier Two Sample Test (C2ST) metric, ROC AUC of classifier between pairs $(x_0, x_1) \sim p_1(x_1)q^\*(x_0 | x_1)$ and $(\hat x_0, x_1) \sim p_1(x_1)q_\theta(x_0 | x_1)$. As the classifier, we used two layer MLP with ReLU activations that takes as input the concatenation of one-hot vectors of $x_0$ and $x_1$. We present C2ST scores in Table 3 of the revised manuscript.
>
> As can be seen from the table, these metrics are not informative. Across all methods the metric values are nearly identical (~$1$), indicating that such a simple classifier is already capable of distinguishing generated samples from real ones. Consequently, as this prevents a meaningful ranking of the methods, we have excluded this experimental setup.

---

> ### Author Response · Authors · 2025-12-03
>
> **[Weakness 3] ([W.4] and [Q.8]) Theoretical Limitation of the Chosen Method: The paper's main innovation appears to be the engineering choice of using a CP decomposition to make the benchmark tractable. However, this choice seems to require the ground-truth problem to be low-rank ($K=4$). Does this not create a fundamental trade-off where the benchmark can be either (a) computable but trivial, or (b) complex but not computable?**
>
> We respectfully disagree. Although choosing a large $K$ is indeed computationally expensive, this does not make the benchmark theoretically intractable. There are no theoretical limitations regarding the value of $K$.
>
> Moreover, we argue that a large $K$ is not required for the benchmark to remain sufficiently challenging or to effectively differentiate strong solvers from weak ones. This is supported by the following observations.
>
> First, CSBM algorithm itself fails to achieve high performance in certain cases (see Table 2), which already indicates that our benchmark is nontrivial even for a neural network-based approach. In particular, one can observe that increasing $N$ (NFE) improves the quality, which is consistent with the typical behavior of diffusion models.
>
> Second, in revised paper (Tables 1, 2, 4, 5) we introduce one intentionally simple baseline solver that serves as sanity check. Specifically, this solver approximates the optimal transport plan $q^*(x_0, x_1)$ by the product of the marginals $p_0(x_0)p_1(x_1)$.
>
> We evaluate this solver in the same way as all other methods, and they consistently obtain very low scores, showing that the benchmark does not assign high metrics to solutions that ignore the SB structure or rely only on feature-wise approximations. While these baselines are not meant to be realistic competitors, together with the performance of nontrivial methods such as CSBM, they indicate that, even without using a large $K$, our benchmark both penalizes naive approximations and has sufficient dynamic range to separate ineffective solvers from effective ones.
>
> Finally, we note that the choice $K=4$ also provides a convenient visualization setup, which enables straightforward visual assessment of the methods' performance.
>
> **[Weakness 4] ([W.5-1] and [Q.5]) "Apples-to-Oranges" Comparison: The evaluation compares DLightSB (a static, dual-potential solver) against CSBM (a dynamic, primal-path solver). Is this not a "apples-to-oranges" comparison? DLightSB solves a simpler, static problem, while CSBM solves a much harder, dynamic one. How can we conclude DLightSB is a better "Schrödinger Bridge solver" from this comparison?**
>
> We emphasize that our goal is to evaluate SB methods themselves rather than a particular formulation, as stated in lines 186-189. We expect that in the future there will be a much wider variety of methods, and our benchmark is designed with this perspective in mind.
>
> To enable a formulation-agnostic evaluation, we avoid relying on benchmark elements specific to the static formulation, such as the potential $v^\*$. Instead, we focus on elements shared across formulations, in particular $q^\*(x_1 | x_0)$, which we compare directly with the methods' conditional distribution $q_\theta(x_1 | x_0)$. Measuring the distance between these two conditionals is sufficient to assess performance, since any SB method ultimately aims to approximate the true conditional $q^\*(x_1 | x_0)$.
>
> **[Weakness 5] ([W.7-1] and [Q.4]) Flawed Evaluation Metrics and Exclusion of Relevant Competitors. The remark in Section 2.4 dismisses "tabular EOT" solvers as non-generative. This seems to overlook the entire class of modern EOT solvers that parameterize the dual potentials ($\phi, \psi$) with neural networks, which are both generative and generalizable. Why were these potential-fitting neural EOT methods—which would provide a true "apples-to-apples" comparison against DLightSB—not included as a baseline?**
>
> Thank you for this suggestion. We agree that this is an interesting direction for the development of discrete solvers. However, we would like to clarify that our intention is not to reject NOT-like methods (e.g., NOT [1], KNOT [2], EgNOT [3], ENOT [4]), but simply not to extend them to discrete data in our paper. The approaches we explicitly reject are discrete EOT methods (including Sinkhorn and others), as detailed in lines 190–196. We believe the three additional methods we propose are adequate for the scope of this work, which is not focused on method development.
>
> [1] Korotin, Alexander, Daniil Selikhanovych, and Evgeny Burnaev. "Neural optimal transport."
>
> [2] Korotin, Alexander, Daniil Selikhanovych, and Evgeny Burnaev. "Kernel neural optimal transport."
>
> [3] Mokrov, Petr, et al. "Energy-guided entropic neural optimal transport."
>
> [4] Gushchin, Nikita, et al. "Entropic neural optimal transport via diffusion processes."

---

> ### Author Response · Authors · 2025-12-03
>
> **[Weakness 6] ([W.7-2] and [Q.6]) Inadequate "Black-Box" Metrics. Why, then, were the evaluation metrics (Shape Score, Trend Score) based on comparing output samples? Why did the authors not use the most direct and rigorous metric: the error between the learned potential and the true potential $v$?**
>
> Recall that our main goal is to construct a formulation-agnostic benchmark as we discuss in [Weakness 4]. Therefore, we do not consider the Shape and Trend metrics inappropriate. Moreover, as we discuss in [Weakness 2], the selected metrics allow us to clearly distinguish poor solvers from strong ones, which we view as sufficient evidence of their suitability.
>
> **Concluding remarks.** Kindly let us know if the clarifications provided address your concerns about our work. We're happy to discuss any remaining questions during the discussion phase. If you find our responses satisfactory, we would appreciate it if you could consider raising your score.

---

### Author Response · Authors · 2025-12-03
**General Answer**

We thank the reviewers for the feedback provided during the review period. After carefully checking the reviewers' comments about our work, we think it is important to clarify the following points:

- **Main Goal of our paper.** All Reviewers raised comments regarding the performance of our DLightSB(-M) methods. We provided more detailed answers to some specific questions from Reviewers in our rebuttal below. However, we want to clarify that these methods are not intended to be state-of-the-art; and that the **main contribution** of our paper is the benchmark itself. The proposed solvers arise as byproducts of our benchmark construction and serve primarily to illustrate and probe the benchmark itself.

- **Inductive bias.** The strong performance of DLightSB(-M) follows from their inductive bias, which arises because they use the same CP-based factorization principles that define our benchmark pairs. Since other solvers do not share this structure, the benchmark offers a **more informative** comparison of their performance. In the **revised manuscript** we describe the impact of the inductive bias more precisely in Lines 505-510 in the Discussion section. Importantly, the benchmark still distinguishes good from poor solvers. The strong results provided by DLightSB(-M) should be interpreted in light of this bias.

- **Additional experiments and results.** Following comments by Reviewer MD4S and ECS7 regarding the complexity of the benchmark. We proposed a **reversed benchmark**. In this setup we considered our mixture of discretized Gaussians as input and a random discretized Gaussian as target. This setup aims to de-bias the DLightSB(-M) methods. However, in this configuration we were only able to compute C2ST, which proved to be not very informative. The **revised manuscript** also adds one additional baseline: we compute both unconditional and conditional metrics on the independent coupling. As shown in Tables 1, 2, 4 and 5, the conditional metrics are more informative, as they cannot be easily exploited by trivial solvers.

---

### Meta-Review · Area_Chair_46KA · 2026-01-03

**Summary:**

The paper proposes a carefully designed benchmark for discrete Schrödinger bridges and entropic optimal transport in high-dimensional discrete spaces, with the key contribution being a tractable synthetic testbed in which ground-truth SB solutions are analytically available through CP-factorized tilting, enabling controlled and reproducible evaluation of discrete SB solvers. While the reviewers raised substantial concerns, particularly regarding inductive bias, baseline coverage, and the scope of evaluation, the rebuttal addresses these issues sufficiently and clarifies the intended contribution of the work. In particular, the authors convincingly reposition DLightSB and its variants as oracle-style reference methods rather than as claims of state-of-the-art performance, which resolves much of the concern about circular evaluation and matched parameterization. They also clearly articulate that the benchmark is formulation-agnostic at the level of conditionals and sample quality, justifying their metric choices and explaining why direct potential comparisons, while possible for some methods, would undermine cross-method comparability. Concerns about limited novelty and benchmark diversity are mitigated by the clarification that discreteness enters through the reference process and dynamics rather than merely through mixture structure, and by the realistic observation that the current ecosystem of discrete SB solvers is small, making the chosen baseline set representative rather than selectively narrow. Several technical confusions (terminology around optimality and projection, simulation-free versus sampling-based solvers, and metric behavior in higher dimensions) were convincingly resolved in the rebuttal, improving the paper’s clarity and positioning. While the benchmark may not yet capture all possible failure modes and richer metrics or harder variants would strengthen it further, the paper succeeds in its stated goal of providing a clean, analytically grounded comparison framework for discrete SB methods. Overall, the rebuttal substantially clarifies scope, limitations, and intent, and the work offers a useful and timely benchmarking contribution that merits acceptance.

**Reviewer Concerns:**

The rebuttal credibly addressed (i) the “circularity” and inductive-bias concern by reframing DLightSB(-M) as oracle baselines and explicitly discussing the bias and its implications, (ii) the “apples-to-oranges” critique by motivating a formulation-agnostic evaluation that compares shared objects (notably conditionals), (iii) several clarity issues (projection/optimality wording, simulation-free vs sampling-based properties, and why some scores behave oddly with dimension), and (iv) the novelty/framing concern by sharpening that the core contribution is the tractable benchmark pipeline rather than Theorem 3.1 itself. Still somewhat outstanding are (a) the desire for broader baseline coverage (especially neural dual-potential EOT-style baselines) and more varied benchmark families beyond CP-friendly constructions, and (b) the worry that current metrics may miss harder failure modes (the “reverse benchmark” attempt did not yield an informative ranking), suggesting room for stronger, more discriminative evaluation in future iterations.

**Reviewer Scores:**

- MD5S (initial score: 2): Likely a small increase, since the authors directly acknowledged the inductive bias, clarified scope (benchmark-first, solvers as byproducts/oracles), and explained why they avoided potential-level metrics for cross-method comparability, but the reviewer’s strongest objections (architectural bias, missing “apples-to-apples” neural potential baselines, and metric adequacy) remain only partially resolved.
- dFbH (initial score: 4): Likely a modest increase, given the clarified role of the reference process in capturing discrete behavior and the explicit acknowledgement that DLightSB performance stems from inductive bias rather than a “best solver” claim, while some novelty concerns (mixture-heavy setup, limited breadth of benchmarks/baselines) would probably still cap enthusiasm.
- ECS7 (initial score: 6): Likely unchanged or slightly higher, since most of their questions were answered (conditional metrics already exist; dimension behavior tied to metric limitations; time-step discussion), while their “benchmark too easy / losing discriminative power” point remains only partly addressed (increasing mixture components not pursued; reverse benchmark uninformative).
- FM4X (initial score: 4): Likely a modest increase, because the authors clarified scope (benchmarking pipeline rather than scalability demonstrations), resolved confusion about “optimal projection,” and answered the simulation-free question, though the reviewer’s desire for broader empirical validation beyond the benchmark would likely keep the score from jumping much further.

---

### Decision · Program_Chairs · 2026-01-26

Accept (Poster)